# Learning to Infer Graphics Programs from Hand-Drawn Images

## Abstract

We introduce a model that learns to convert simple hand drawings into graphics programs written in a subset of LaTeX. The model combines techniques from deep learning and program synthesis. We learn a convolutional neural network that proposes plausible drawing primitives that explain an image. These drawing primitives are like a trace of the set of primitive commands issued by a graphics program. We learn a model that uses program synthesis techniques to recover a graphics program from that trace. These programs have constructs like variable bindings, iterative loops, or simple kinds of conditionals. With a graphics program in hand, we can correct errors made by the deep network and extrapolate drawings. Taken together these results are a step towards agents that induce useful, human-readable programs from perceptual input.

## 1 Introduction

How can an agent convert noisy, high-dimensional perceptual input to a symbolic, abstract object, such as a computer program? Here we consider this problem within a graphics program synthesis domain. We develop an approach for converting hand drawings into executable source code for drawing the original image. The graphics programs in our domain draw simple figures like those found in machine learning papers (see Fig. 1a).

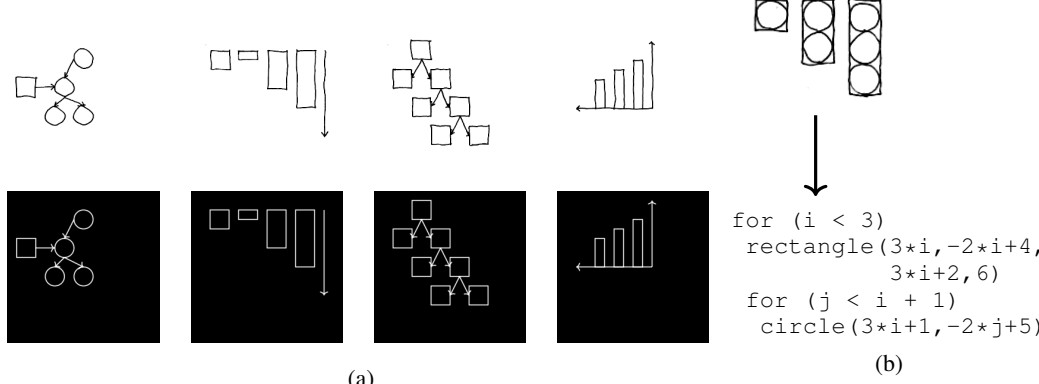

```
for (i < 3)
  rectangle(3*i,-2*i+4,
            3*i+2,6)
  for (j < i + 1)
    circle(3*i+1,-2*j+5)
```

(a)                                                      (b)

Figure 1: (a): Model learns to convert hand drawings (top) into LaTeX (rendered below). (b): Synthesizes high-level *graphics program* from hand drawing.

The key observation behind our work is that generating a programmatic representation from an image of a diagram involves two distinct steps that require different technical approaches. The first step involves identifying the components such as rectangles, lines and arrows that make up the image. The second step involves identifying the high-level structure in how the components were drawn. In Fig. 1(b), it means identifying a pattern in how the circles and rectangles are being drawn that is best described with two nested loops, and which can easily be extrapolated to a bigger diagram.

We present a hybrid architecture for inferring graphics programs that is structured around these two steps. For the first step, a deep network to infers a set of primitive shape-drawing commands. We refer

to this set as a *trace set* since it corresponds to the set of commands in a program's execution trace but lacks the high-level structure that determines how the program decided to issue them. The second step involves synthesizing a high-level program capable of producing the trace set identified by the first phase. We achieve this by *constraint-based program synthesis* (Solar Lezama, 2008). The program synthesizer searches the space of possible programs for one capable of producing the desired trace set – inducing structures like symmetries, loops, or conditionals. Although these program synthesizers do not need any training data, we show how to learn a *search policy* in order to synthesize programs an order of magnitude faster than constraint-based synthesis techniques alone.

One might be tempted to try to go directly from perceptual input (an image) to a program, much like recent neural models of program induction that regress from problem statements to programs (e.g., Devlin et al. (2017)). We advocate an alternative framing, which we call The Trace Hypothesis:

**The Trace Hypothesis.** The set of commands issued by a program, which we call a *trace set*, is the correct liaison between high-dimensional unstructured perceptual input and high-level structured symbolic representations.

The roadmap of our paper is structured around the trace hypothesis, as outlined in Fig. 2.

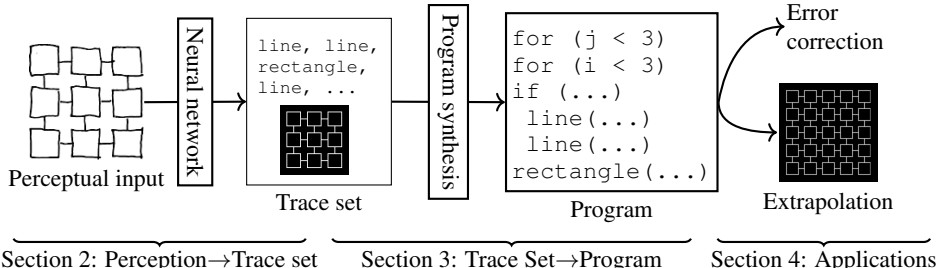

Figure 2: Both the paper and the system pipeline are structured around the *trace hypothesis*

The new contributions of this work are: (1) The trace hypothesis: a framework for going from perception to programs, which connects this work to other trace–based models, like the Neural Program Interpreter (Reed & de Freitas, 2015); (2) A model based on the trace hypothesis that converts sketches to high-level programs: in contrast to converting images to vectors or low-level parses (Huang et al., 2017; Nishida et al., 2016; Wu et al., 2017; Beltramelli, 2017; Deng et al., 2017). (3) A generic algorithm for learning a policy for efficiently searching for programs, building on Levin search (Levin, 1973) and recent work like DeepCoder (Balog et al., 2016).

Even with the high-level idea of a trace set, going from hand drawings to programs remains difficult. We address these challenges: (1) Inferring trace sets from images requires *domain-specific* design choices from the deep learning and computer vision toolkits (Sec. 2). (2) Generalizing to noisy hand drawings, we will show, requires learning a *domain-specific* noise model that is invariant to the variations across hand drawings (Sec. 2.1). (3) Discovering good programs requires solving a difficult combinatorial search problem, because the programs are often long and complicated (e.g., 9 lines of code, with nested loops and conditionals). We give a *domain-general* framework for learning a search policy that quickly guides program synthesizers toward the target programs (Sec. 3.1).

## 2  NEURAL ARCHITECTURE FOR INFERRING TRACE SETS

We developed a deep network architecture for efficiently inferring a trace set, $T$, from an image, $I$. Our model combines ideas from Neurally-Guided Procedural Models (Ritchie et al., 2016) and Attend-Infer-Repeat (Eslami et al., 2016). The network constructs the trace set one drawing command at a time, conditioned on what it has drawn so far. Fig. 3 illustrates this architecture. We first pass a $256 \times 256$ target image and a rendering of the trace set so far (encoded as a two-channel image) to a convolutional network. Given the features extracted by the convnet, a multilayer perceptron then predicts a distribution over the next drawing command to add to the trace set (see Tbl. 1). We also use a differentiable attention mechanism (Spatial Transformer Networks: Jaderberg et al. (2015)) to let

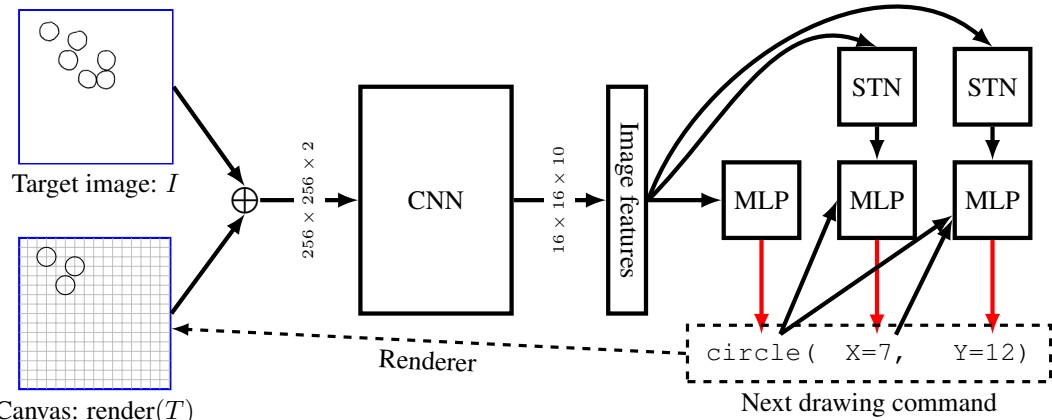

Figure 3: Our neural architecture for inferring the trace set of a graphics program from its output. Blue: network inputs. Black: network operations. Red: samples from a multinomial. Typewriter font: network outputs. Renders snapped to a $16 \times 16$ grid, illustrated in gray. STN (spatial transformer network) is a differentiable attention mechanism (Jaderberg et al., 2015).

Table 1: Primitive drawing commands currently supported by our model.

| | |
|---|---|
| `circle`$(x, y)$ | Circle at $(x, y)$ |
| `rectangle`$(x_1, y_1, x_2, y_2)$ | Rectangle with corners at $(x_1, y_1)$ & $(x_2, y_2)$ |
| `line`$(x_1, y_1, x_2, y_2,$ | Line from $(x_1, y_1)$ to $(x_2, y_2)$, |
| $\quad$ arrow $\in \{0, 1\}$, dashed $\in \{0, 1\})$ | $\quad$ optionally with an arrow and/or dashed |
| `STOP` | Finishes trace set inference |

the model attend to different regions of the image while predicting drawing commands. We currently constrain coordinates to lie on a discrete $16 \times 16$ grid, but the grid could be made arbitrarily fine.

We train the network by sampling trace sets $T$ and target images $I$ for randomly generated scenes and maximizing the likelihood of $T$ given $I$ with respect to the model parameters, $\theta$, by gradient ascent. We trained the network on $10^5$ scenes, which takes a day on an Nvidia TitanX GPU.

Our network can "derender" random synthetic images by doing a beam search to recover trace sets maximizing $\mathbb{P}_\theta[T|I]$. But, if the network predicts an incorrect drawing command, it has no way of recovering from that error. For added robustness we treat the network outputs as proposals for a Sequential Monte Carlo (SMC) sampling scheme (Doucet et al., 2001). The SMC sampler is designed to sample from the distribution $\propto L(I|\text{render}(T))\mathbb{P}_\theta[T|I]$, where $L(\cdot|\cdot)$ uses the pixel-wise distance between two images as a proxy for a likelihood. Here, the network is learning a proposal distribution in an amortized way (Paige & Wood, 2016) and using it to invert a generative model (the renderer).

**Experiment 1: Figure 4.** To evaluate which components of the model are nec-

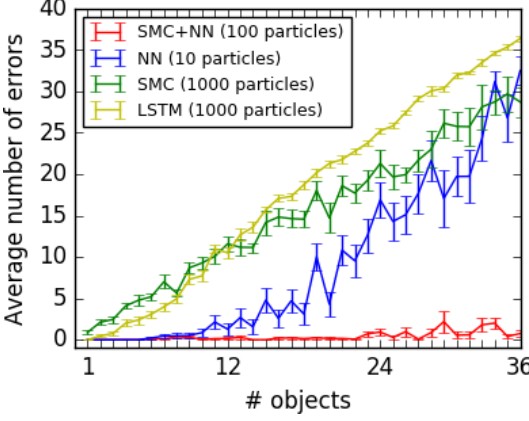

Figure 4: Parsing LaTeX output after training on diagrams with $\leq 12$ objects. Model generalizes to scenes with many more objects. Neither SMC nor the neural network are sufficient on their own. # particles varies by model: we compare the models *with equal runtime* ($\approx 1$ sec/object)

essary to parse complicated scenes, we compared the neural network with SMC against the neural network by itself or SMC by itself. Only the combination of the two passes a critical test of generalization: when trained on images with $\leq 12$ objects, it successfully parses scenes with many more objects than the training data. We compare with a baseline that produces the trace set in one shot by using the CNN to extract features of the input which are passed to an LSTM which finally predicts the trace set token-by-token (LSTM in Fig. 4). This architecture is used in several successful neural models of image captioning (e.g., Vinyals et al. (2015)), but, for this domain, cannot parse cluttered scenes with many objects.

## 2.1 GENERALIZING TO REAL HAND DRAWINGS

We trained the model to generalize to hand drawings by introducing noise into the renderings of the training target images. We designed this noise process to introduce the kinds of variations found in hand drawings (see supplement for details).

Our neurally-guided SMC procedure used pixel-wise distance as a surrogate for a likelihood function ($L(\cdot|\cdot)$ in section 2). But pixel-wise distance fares poorly on hand drawings, which never exactly match the model's renders. So, for hand drawings, we learn a surrogate likelihood function, $L_{\text{learned}}(\cdot|\cdot)$. The density $L_{\text{learned}}(\cdot|\cdot)$ is predicted by a convolutional network that we train to predict the distance between two trace sets conditioned upon their renderings. We train our likelihood surrogate to approximate the symmetric difference, which is the number of drawing commands by which two trace sets differ:

$$-\log L_{\text{learned}}(\text{render}(T_1)|\text{render}(T_2)) \approx |T_1 - T_2| + |T_2 - T_1| \tag{1}$$

**Experiment 2: Figures 5–7.** We evaluated, but did not train, our system on 100 real hand-drawn figures; see Fig. 5–6. These were drawn carefully but not perfectly with the aid of graph paper. For each drawing we annotated a ground truth trace set and had the neurally guided SMC sampler produce $10^3$ samples. For 63% of the drawings, the Top-1 most likely sample exactly matches the ground truth; with more samples, the model finds trace sets that are closer to the ground truth annotation (Fig. 7). We will show that the program synthesizer corrects some of these small errors (Sec. 4.1).

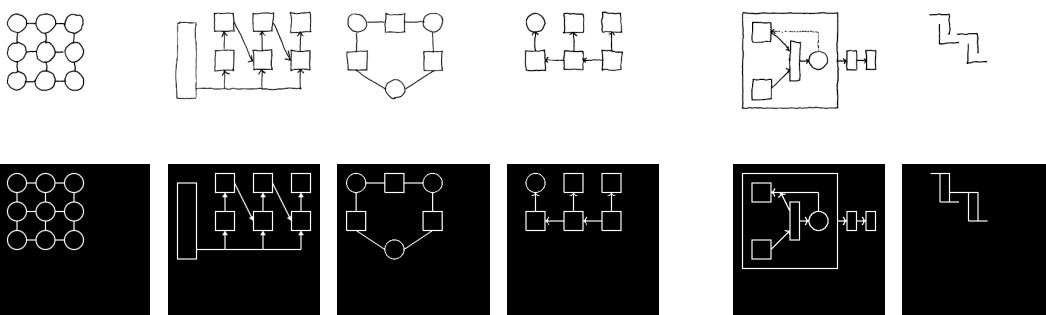

Figure 5: Left to right: Ising model, recurrent network architecture, figure from a deep learning textbook Goodfellow et al. (2016), graphical model

Figure 6: Near misses. Rightmost: illusory contours (note: no SMC)

## 3 SYNTHESIZING GRAPHICS PROGRAMS FROM TRACE SETS

Although the trace set of a graphics program describes the contents of a scene, it does not encode higher-level features of the image, such as repeated motifs or symmetries. A *graphics program* better describes such structures. We seek to synthesize graphics programs from their trace sets.

We constrain the space of programs by writing down a context free grammar over programs – what in the program languages community is called a Domain Specific Language (DSL) (Polozov & Gulwani, 2015). Our DSL (Tbl. 2) encodes prior knowledge of what graphics programs tend to look like.

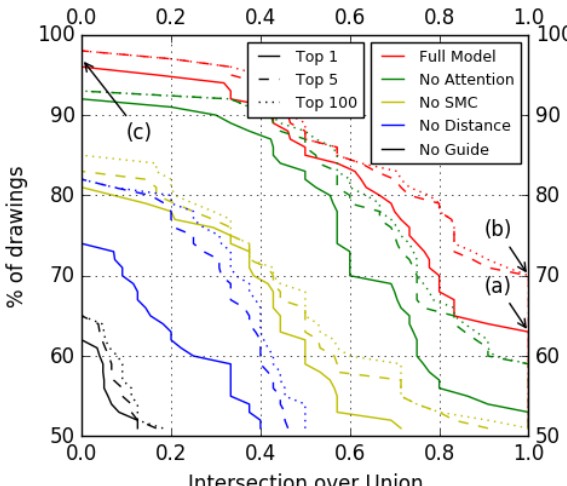

Figure 7: How close are the model's outputs to the ground truth on hand drawings, as we consider larger sets of samples (1, 5, 100)? Distance to ground truth trace set measured by the intersection over union (IoU) of predicted vs. ground truth: IoU of sets $A$ and $B$ is $|A \cap B|/|A \cup B|$. (a) for 63% of drawings the model's top prediction is exactly correct; (b) for 70% of drawings the ground truth is in the top 5 model predictions; (c) for 4% of drawings all of the model outputs have no overlap with the ground truth. Red: the full model. Other colors: lesioned versions of our model.

Table 2: Grammar over graphics programs. We allow loops (`for`) with conditionals (`if`), vertical/horizontal reflections (`reflect`), variables (Var) and affine transformations ($\mathbb{Z} \times$ Var$+\mathbb{Z}$).

| | |
|---|---|
| Program→ | Statement; $\cdots$; Statement |
| Statement→ | `circle`(Expression,Expression) |
| Statement→ | `rectangle`(Expression,Expression,Expression,Expression) |
| Statement→ | `line`(Expression,Expression,Expression,Expression,Boolean,Boolean) |
| Statement→ | `for`$(0 \leq$ Var $<$ Expression$)$ { `if` (Var $> 0$) { Program }; Program } |
| Statement→ | `reflect`(Axis) { Program } |
| Expression→ | $\mathbb{Z} \times$Var$+\mathbb{Z}$ |
| Axis→ | `X = `$\mathbb{Z}$`|Y = `$\mathbb{Z}$ |
| $\mathbb{Z} \to$ | an integer |

Given the DSL and a trace set $T$, we want a program that both evaluates to $T$ and, at the same time, is the "best" explanation of $T$. For example, we might prefer more general programs or, in the spirit of Occam's razor, prefer shorter programs. We wrap these intuitions up into a cost function over programs, and seek the minimum cost program consistent with $T$:

$$\text{program}(T) = \underset{p \in \text{DSL, s.t. } p \text{ evaluates to } T}{\arg \min} \text{cost}(p) \qquad (2)$$

We define the cost of a program to be the number of Statement's it contains (Tbl. 2). We also penalize using many different numerical constants; see supplement.

The constrained optimization problem in Eq. 2 is intractable in general, but there exist efficient-in-practice tools for finding exact solutions to such program synthesis problems. We use the state-of-the-art Sketch tool (Solar Lezama, 2008). Sketch takes as input a space of programs, along with a specification of the program's behavior and optionally a cost function. It translates the synthesis problem into a constraint satisfaction problem and then uses a SAT solver to find a minimum-cost program satisfying the specification. Sketch requires a *finite program space*, which here means that the depth of the program syntax tree is bounded (we set the bound to 3), but has the guarantee that it always eventually finds a globally optimal solution. In exchange for this optimality guarantee it comes with no guarantees on runtime. For our domain synthesis times vary from minutes to hours, with 27% of the drawings timing out the synthesizer after 1 hour. Tbl. 3 shows programs recovered by our system. A main impediment to our use of these general techniques is the prohibitively high cost of searching for programs. We next describe how to learn to synthesize programs much faster (Sec. 3.1), timing out on 2% of the drawings and solving 58% of problems within a minute.

Table 3: Example drawings (left), their ground truth trace sets (middle left), and programs synthesized from these trace sets (middle right). Compared to the trace sets the programs are more compressive (right: programs have fewer lines than traces) and automatically group together related drawing commands. Note the nested loops and special case conditionals in the Ising model, combination of symmetry and iteration in the bottom figure, affine transformations in the top figure, and the complicated program in the second figure to bottom.

| Drawing | Trace Set | Program | Compression factor |
|---|---|---|---|
| | ```Line(2,15, 4,15)```
```Line(4,9, 4,13)```
```Line(3,11, 3,14)```
```Line(2,13, 2,15)```
```Line(3,14, 6,14)```
```Line(4,13, 8,13)``` | ```for(i<3)```
  ```line(i,-1*i+6,```
     ```2*i+2,-1*i+6)```
  ```line(i,-2*i+4,i,-1*i+6)``` | $\frac{6}{3} = 2\text{x}$ |
| | ```Circle(5,8)```
```Circle(2,8)```
```Circle(8,11)```
```Line(2,9, 2,10)```
```Circle(8,8)```
```Line(3,8, 4,8)```
```Line(3,11, 4,11)```
*... etc. ...; 21 lines* | ```for(i<3)```
  ```for(j<3)```
   ```if(j>0)```
    ```line(-3*j+8,-3*i+7,```
      ```-3*j+9,-3*i+7)```
    ```line(-3*i+7,-3*j+8,```
      ```-3*i+7,-3*j+9)```
   ```circle(-3*j+7,-3*i+7)``` | $\frac{21}{6} = 3.5\text{x}$ |
| | ```Rectangle(1,10,3,11)```
```Rectangle(1,12,3,13)```
```Rectangle(4,8,6,9)```
```Rectangle(4,10,6,11)```
*... etc. ...; 16 lines* | ```for(i<4)```
  ```for(j<4)```
   ```rectangle(-3*i+9,-2*j+6,```
      ```-3*i+11,-2*j+7)``` | $\frac{16}{3} = 5.3\text{x}$ |
| | ```Line(3,10,3,14,arrow)```
```Rectangle(11,8,15,10)```
```Rectangle(11,14,15,15)```
```Line(13,10,13,14,arrow)```
*... etc. ...; 16 lines* | ```for(i<3)```
  ```line(7,1,5*i+2,3,arrow)```
  ```for(j<i+1)```
   ```if(j>0)```
    ```line(5*j-1,9,5*i,5,arrow)```
   ```line(5*j+2,5,5*j+2,9,arrow)```
  ```rectangle(5*i,3,5*i+4,5)```
  ```rectangle(5*i,9,5*i+4,10)```
```rectangle(2,0,12,1)``` | $\frac{16}{9} = 1.8\text{x}$ |
| | ```Circle(2,8)```
```Rectangle(6,9, 7,10)```
```Circle(8,8)```
```Rectangle(6,12, 7,13)```
```Rectangle(3,9, 4,10)```
*... etc. ...; 9 lines* | ```reflect(y=8)```
  ```for(i<3)```
   ```if(i>0)```
    ```rectangle(3*i-1,2,3*i,3)```
   ```circle(3*i+1,3*i+1)``` | $\frac{9}{5} = 1.8\text{x}$ |

## 3.1 LEARNING A SEARCH POLICY FOR SYNTHESIZING PROGRAMS

We want to leverage powerful, domain-general techniques from the program synthesis community, but make them much faster by learning a domain-specific *search policy*. A search policy poses search problems like those in Eq. 2, but also offers additional constraints on the structure of the program (Tbl. 4). For example, a policy might decide to first try searching over small programs before searching over large programs, or decide to prioritize searching over programs that have loops.

A search policy $\pi_\theta(\sigma|T)$ takes as input a trace set $T$ and predicts a distribution over synthesis problems, each of which is written $\sigma$ and corresponds to a set of possible programs to search over (so $\sigma \subseteq$ DSL). Good policies will prefer tractable program spaces, so that the search procedure will terminate early, but should also prefer program spaces likely to contain programs that concisely explain the data. These two desiderata are in tension: tractable synthesis problems involve searching over smaller spaces, but smaller spaces are less likely to contain good programs. Our goal now is to find the parameters of the policy, written $\theta$, which best navigate this trade-off.

Given a search policy, what is the best way of using it to quickly find minimum cost programs? We use a *bias-optimal search algorithm* (Schmidhuber, 2004):

**Definition: Bias-optimality.** A search algorithm is *n-bias optimal* with respect to a distribution $\mathbb{P}_{\text{bias}}[\cdot]$ if it is guaranteed to find a solution in $\sigma$ after searching for at least time $n \times \frac{t(\sigma)}{\mathbb{P}_{\text{bias}}[\sigma]}$, where $t(\sigma)$ is the time it takes to verify that $\sigma$ contains a solution to the search problem.

An example of a 1-bias optimal search algorithm is a time-sharing system that allocates $\mathbb{P}_{\text{bias}}[\sigma]$ of its time to trying $\sigma$. We construct a 1-bias optimal search algorithm by identifying $\mathbb{P}_{\text{bias}}[\sigma] = \pi_\theta(\sigma|T)$ and $t(\sigma) = t(\sigma|T)$, where $t(\sigma|T)$ is how long the synthesizer takes to search $\sigma$ for a program for $T$. This means that the search algorithm explores the entire program space, but spends most of its time in the regions of the space that the policy judges to be most promising.

Now in theory any $\pi_\theta(\cdot|\cdot)$ is a bias-optimal searcher. But the actual runtime of the algorithm depends strongly upon the bias $\mathbb{P}_{\text{bias}}[\cdot]$. Our new approach is to learn $\mathbb{P}_{\text{bias}}[\cdot]$ by picking the policy minimizing the expected bias-optimal time to solve a training corpus, $\mathcal{D}$, of graphics program synthesis problems:

$$\text{Loss}(\theta; \mathcal{D}) = \mathbb{E}_{T \sim \mathcal{D}} \left[ \min_{\sigma \in \text{BEST}(T)} \frac{t(\sigma|T)}{\pi_\theta(\sigma|T)} \right] + \lambda \|\theta\|_2^2 \qquad (3)$$

where $\sigma \in \text{BEST}(T)$ if a minimum cost program for $T$ is in $\sigma$.

Practically, bias optimality has now bought us the following: (1) a guarantee that the policy will always find the minimum cost program; and (2) a differentiable loss function for the policy parameters that takes into account the cost of searching, in contrast to e.g. DeepCoder (Balog et al., 2016).

To generate a training corpus for learning a policy which minimizes this loss, we synthesized minimum cost programs for each trace set of our hand drawings and for each $\sigma$. We locally minimize this loss using gradient descent. Because we want to learn a policy from only 100 hand-drawn diagrams, we chose a simple low-capacity, bilinear model for a policy:

$$\pi_\theta(\sigma|T) \propto \exp\left(\phi_{\text{params}}(\sigma)^\top \theta \phi_{\text{trace}}(T)\right) \qquad (4)$$

where $\phi_{\text{params}}(\sigma)$ is a one-hot encoding of the parameter settings of $\sigma$ (see Tbl. 4) and $\phi_{\text{trace}}(T)$ extracts a few simple features of the trace set $T$; see supplement for details.

**Experiment 3: Figure 8.** We compare synthesis times for our learned search policy with two alternatives: *Sketch*, which poses the entire problem wholesale to the Sketch program synthesizer; and an *Oracle*, a policy which always picks the quickest to search $\sigma$ also containing a minimum cost program. Our approach improves upon Sketch by itself, and comes close to the Oracle's performance. One could never construct this Oracle, because the agent does not know ahead of time which $\sigma$'s contain minimum cost programs nor does it know how long each $\sigma$ will take to search. With this learned policy in hand we can synthesize 58% of programs within a minute.

Table 4: Parameterization of different ways of posing the program synthesis problem. The policy learns to choose parameters likely to quickly yield a minimal cost program. We slightly abuse notation by writing $\sigma$ to mean an assignment to each of these parameters (so $\sigma$ assumes one of 24 different values) and to mean the set of programs selected by that parameterization (so $\sigma \subseteq \text{DSL}$)

| Parameter | Description | Range |
|---|---|---|
| Loops? | Is the program allowed to loop? | $\{\text{True}, \text{False}\}$ |
| Reflects? | Is the program allowed to have reflections? | $\{\text{True}, \text{False}\}$ |
| Incremental? | Solve the problem piece-by-piece or all at once? | $\{\text{True}, \text{False}\}$ |
| Maximum depth | Bound on the depth of the program syntax tree | $\{1, 2, 3\}$ |

## 4 APPLICATIONS OF GRAPHICS PROGRAM SYNTHESIS

Why synthesize a graphics program, if the trace set already suffices to recover the objects in an image? Within our domain of hand-drawn figures, graphics program synthesis has several uses:

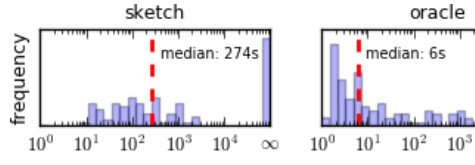 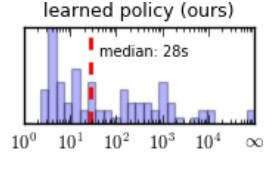

Figure 8: How long does it typically take to synthesize a minimum cost program? Sketch: out-of-the-box performance of the Sketch (Solar Lezama, 2008) program synthesizer. Oracle: upper bounds the performance of any search policy. Learned policy: a bias-optimal learned search policy running on an ideal timesharing machine. $\infty$ = timeout. Red dashed line is median time. Learned policy evaluated using 20-fold cross validation.

## 4.1 Correcting errors made by the neural network

The program synthesizer corrects errors made by the neural network by favoring trace sets which lead to more concise or general programs. For example, figures with perfectly aligned objects are preferable, and precise alignment lends itself to short programs. Concretely, we run the program synthesizer on the Top-$k$ most likely trace sets output by the neurally guided sampler. Then, the system reranks the Top-$k$ by the prior probability of their programs. The prior probability of a program is learned by picking the prior maximizing the likelihood of the ground truth trace sets; see supplement for details. But, this procedure can only correct errors when a correct trace set is in the Top-$k$. Our sampler could only do better on 7/100 drawings by looking at the Top-100 samples (see Fig. 7), precluding a statistically significant analysis of how much learning a prior over programs could help correct errors. But, learning this prior does sometimes help correct mistakes made by the neural network; see Fig. 9 for a representative example of the kinds of corrections that it makes. See supplement for details.

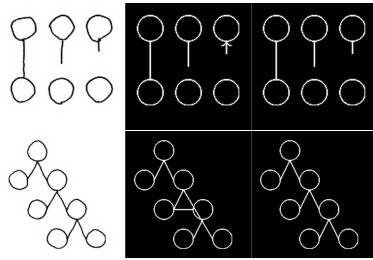

Figure 9: Left: hand drawings. Center: interpretations favored by the deep network. Right: interpretations favored after learning a prior over programs. The prior favors simpler programs, thus (top) continuing the pattern of not having an arrow is preferred, or (bottom) continuing the "binary search tree" is preferred.

## 4.2 Extrapolating figures

Having access to the source code of a graphics program facilitates coherent, high-level image editing. For example we can extrapolate figures by increasing the number of times that loops are executed. Extrapolating repetitive visuals patterns comes naturally to humans, and is a practical application: imagine hand drawing a repetitive graphical model structure and having our system automatically induce and extend the pattern. Fig. 10 shows extrapolations produced by our system.

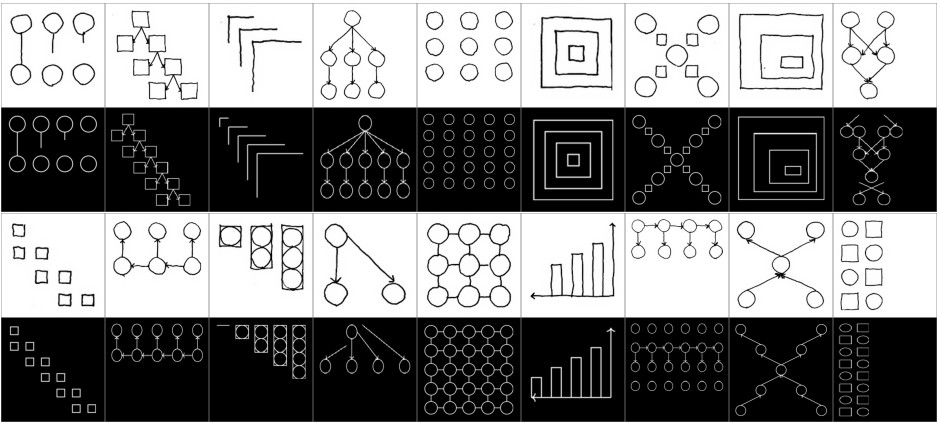

Figure 10: Top, white: hand drawings. Bottom, black: extrapolations produced by our system.

## 5 RELATED WORK

**Program Induction:** Our approach to learning to search for programs draws theoretical underpinnings from Levin search (Levin, 1973; Solomonoff, 1984) and Schmidhuber's OOPS model (Schmidhuber, 2004). DeepCoder (Balog et al., 2016) is a recent model which, like ours, learns to predict likely program components. Our work differs because we treat the problem as *metareasoning*, identifying and modeling the trade-off between tractability and probability of success. TerpreT (Gaunt et al., 2016) systematically compares constraint-based program synthesis techniques against gradient-based search techniques, like those used to train Differentiable Neural Computers (Graves et al., 2016). The TerpreT experiments motivate our use of constraint-based techniques.

**Deep Learning:** Our neural network bears resemblance to the Attend-Infer-Repeat (AIR) system, which learns to decompose an image into its constituent objects (Eslami et al., 2016). AIR learns an iterative inference scheme which infers objects one by one and also decides when to stop inference. Our network differs in its architecture and training regime: AIR learns a recurrent auto-encoding model via variational inference, whereas our parsing stage learns an autoregressive-style model from randomly-generated (trace, image) pairs.

IM2LATEX (Deng et al., 2017) is a recent work that also converts images to LATEX. Their goal is to derender LATEX equations, which recovers a markup language representation. Our goal is to go from noisy input to a high-level program, which goes beyond markup languages by supporting programming constructs like loops and conditionals. Recovering a high-level program is more challenging than recovering markup because it is a highly under constrained symbolic reasoning problem.

Our image-to-trace parsing architecture builds on prior work on controlling procedural graphics programs (Ritchie et al., 2016). We adapt this method to a different visual domain (figures composed of multiple objects), using a broad prior over possible scenes as the initial program and viewing the trace through the guide program as a symbolic parse of the target image. We then show how to efficiently synthesize higher-level programs from these traces.

In the computer graphics literature, there have been other systems which convert sketches into procedural representations. One uses a convolutional network to match a sketch to the output of a parametric 3D modeling system (Huang et al., 2017). Another uses convolutional networks to support sketch-based instantiation of procedural primitives within an interactive architectural modeling system (Nishida et al., 2016). Both systems focus on inferring fixed-dimensional parameter vectors. In contrast, we seek to automatically infer a structured, programmatic representation of a sketch which captures higher-level visual patterns.

**Hand-drawn sketches:** Prior work has also applied sketch-based program synthesis to authoring graphics programs. Sketch-n-Sketch is a bi-directional editing system in which direct manipulations to a program's output automatically propagate to the program source code (Hempel & Chugh, 2016). We see this work as complementary to our own: programs produced by our method could be provided to a Sketch-n-Sketch-like system as a starting point for further editing.

The CogSketch system (Forbus et al., 2011) also aims to have a high-level understanding of hand-drawn figures. Their primary goal is cognitive modeling (they apply their system to solving IQ-test style visual reasoning problems), whereas we are interested in building an automated AI application (e.g. in our system the user need not annotate which strokes correspond to which shapes; our neural network produces something equivalent to the annotations).

**The Trace Hypothesis:** The idea that an execution trace could assist in program learning goes back to the 1970's (Summers, 1977) and has been applied in neural models of program induction, like Neural Program Interpreters (Reed & de Freitas, 2015), or DeepCoder, which predicts what functions occur in the execution trace (Balog et al., 2016). Our contribution to this idea is the trace hypothesis: that trace sets can be inferred from perceptual data, and that the trace set is a useful bridge between perception and symbolic representation. Our work is the first to articulate and explore this hypothesis by demonstrating how a trace could be inferred and how it can be used to synthesize a high-level program.

## 6 CONTRIBUTIONS

We have presented a system for inferring graphics programs which generate LaTeX-style figures from hand-drawn images. The system uses a combination of deep neural networks and stochastic search to parse drawings into symbolic trace sets; it then feeds these traces to a general-purpose program synthesis engine to infer a structured graphics program. We evaluated our model's performance at parsing novel images, and we demonstrated its ability to extrapolate from provided drawings.

In the near future, we believe it will be possible to produce professional-looking figures just by drawing them and then letting an artificially-intelligent agent write the code. More generally, we believe the trace hypothesis, as realized in our two-phase system—parsing into trace sets, then searching for a low-cost symbolic program which generates those traces—may be a useful paradigm for other domains in which agents must programmatically reason about noisy perceptual input.

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

# SUPPLEMENT TO: LEARNING TO INFER GRAPHICS PROGRAMS FROM HAND-DRAWN IMAGES

**Anonymous authors**

## 1 CORRECTING ERRORS MADE BY THE NEURAL NETWORK

The program synthesizer can help correct errors from the execution trace proposal network by favoring execution traces which lead to more concise or general programs. For example, one generally prefers figures with perfectly aligned objects over figures whose parts are slightly misaligned – and precise alignment lends itself to short programs. Similarly, figures often have repeated parts, which the program synthesizer might be able to model as a loop or reflectional symmetry. So, in considering several candidate traces proposed by the neural network, we might prefer traces whose best programs have desirable features such being short or having iterated structures.

Concretely, we implemented the following scheme: for an image $I$, the neurally guided sampling scheme of section 3 of the main paper samples a set of candidate traces, written $\mathcal{F}(I)$. Instead of predicting the most likely trace in $\mathcal{F}(I)$ according to the neural network, we can take into account the programs that best explain the traces. Writing $\hat{T}(I)$ for the trace the model predicts for image $I$,

$$\hat{T}(I) = \arg\max_{T \in \mathcal{F}(I)} L_{\text{learned}}(I|\text{render}(T)) \times \mathbb{P}_\theta[T|I] \times \mathbb{P}_\beta[\text{program}(T)] \tag{1}$$

where $\mathbb{P}_\beta[\cdot]$ is a prior probability distribution over programs parameterized by $\beta$. This is equivalent to doing MAP inference in a generative model where the program is first drawn from $\mathbb{P}_\beta[\cdot]$, then the program is executed deterministically, and then we observe a noisy version of the program's output, where $L_{\text{learned}}(I|\text{render}(\cdot)) \times \mathbb{P}_\theta[\cdot|I]$ is our observation model.

Given a corpus of graphics program synthesis problems with annotated ground truth traces (i.e. $(I, T)$ pairs), we find a maximum likelihood estimate of $\beta$:

$$\beta^* = \arg\max_\beta \mathbb{E}\left[\log \frac{\mathbb{P}_\beta[\text{program}(T)] \times L_{\text{learned}}(I|\text{render}(T)) \times \mathbb{P}_\theta[T|I]}{\sum_{T' \in \mathcal{F}(I)} \mathbb{P}_\beta[\text{program}(T')] \times L_{\text{learned}}(I|\text{render}(T')) \times \mathbb{P}_\theta[T'|I]}\right] \tag{2}$$

where the expectation is taken both over the model predictions and the $(I, T)$ pairs in the training corpus. We define $\mathbb{P}_\beta[\cdot]$ to be a log linear distribution $\propto \exp(\beta \cdot \phi(\text{program}))$, where $\phi(\cdot)$ is a feature extractor for programs. We extract a few basic features of a program, such as its size and how many loops it has, and use these features to help predict whether a trace is the correct explanation for an image.

We synthesized programs for the top 10 traces output by the deep network. Learning this prior over programs can help correct mistakes made by the neural network, and also occasionally introduces mistakes of its own; see Fig. 1 for a representative example of the kinds of corrections that it makes. On the whole it modestly improves our Top-1 accuracy from 63% to 67%. Recall that from Fig. 6 of the main paper that the best improvement in accuracy we could possibly get is 70% by looking at the top 10 traces.

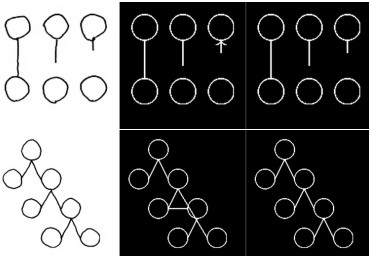

Figure 1: Left: hand drawing. Center: interpretation favored by the deep network. Right: interpretation favored after learning a prior over programs. Our learned prior favors shorter, simpler programs, thus (top example) continuing the pattern of not having an arrow is preferred, or (bottom example) continuing the binary search tree is preferred.

## 2 MEASURING SIMILARITY BETWEEN DRAWINGS

We measure the similarity between two drawings by extracting features of the best programs that describe them. Our features are counts of the number of times that different components in the DSL were used. We project these features down to a 2-dimensional subspace using primary component analysis (PCA); see Fig.2. One could use many alternative similarity metrics between drawings which would capture pixel-level similarities while missing high-level geometric similarities. We used our learned distance metric between traces, $L_{\text{learned}}(\cdot|\cdot)$, and projected to a 2-dimensional subspace using multidimensional scaling (MDS: (1)). This reveals similarities between the objects in the drawings, while missing similarities at the level of the program.

## 3 LEARNING A BIAS OPTIMAL POLICY

Recall from the main paper that our goal is to estimate the policy minimizing the following loss:

$$\text{LOSS}(\theta; \mathcal{D}) = \mathbb{E}_{T \sim \mathcal{D}} \left[ \min_{\sigma \in \text{BEST}(T)} \frac{t(\sigma|T)}{\pi_\theta(\sigma|T)} \right] + \lambda \|\theta\|_2^2 \qquad (3)$$

where $\sigma \in \text{BEST}(T)$ if a minimum cost program for $T$ is in $\sigma$.

We make this optimization problem tractable by annealing our loss function during gradient descent:

$$\text{LOSS}_\beta(\theta; \mathcal{D}) = \mathbb{E}_{T \sim \mathcal{D}} \left[ \text{SOFTMINIMUM}_\beta \left\{ \frac{t(\sigma|T)}{\pi_\theta(\sigma|T)} : \sigma \in \text{BEST}(T) \right\} \right] + \lambda \|\theta\|_2^2 \qquad (4)$$

$$\text{where SOFTMINIMUM}_\beta(x_1, x_2, x_3, \cdots) = \sum_n x_n \frac{e^{-\beta x_n}}{\sum_{n'} e^{-\beta x_{n'}}} \qquad (5)$$

Notice that $\text{SOFTMINIMUM}_{\beta=\infty}(\cdot)$ is just $\min(\cdot)$. We set the regularization coefficient $\lambda = 0.1$ and minimize equation 4 using Adam for 2000 steps, linearly increasing $\beta$ from 1 to 2.

We parameterize the space of policies as a simple log bilinear model:

$$\pi_\theta(\sigma|T) \propto \exp\left(\phi_{\text{params}}(\sigma)^\top \theta \phi_{\text{trace}}(T)\right) \qquad (6)$$

where:

$$\phi_{\text{params}}(\sigma) = [\mathbb{1}[\sigma \text{ can loop}];$$
$$\mathbb{1}[\sigma \text{ can reflect}];$$
$$\mathbb{1}[\sigma \text{ is incremental}];$$
$$\mathbb{1}[\sigma \text{ has depth bound 1}]; \mathbb{1}[\sigma \text{ has depth bound 2}]; \mathbb{1}[\sigma \text{ has depth bound 3}];]$$
$$\phi_{\text{trace}}(T) = [\# \text{ circles in } T; \# \text{ rectangles in } T; \# \text{ lines in } T; 1]$$

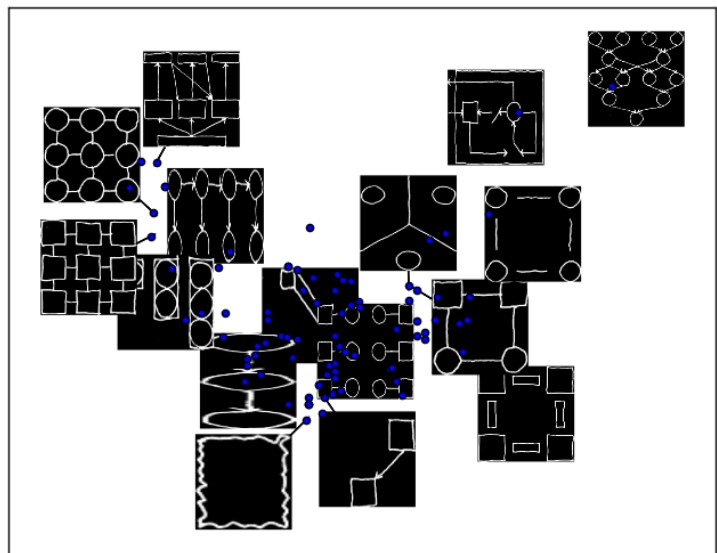

Figure 2: PCA on features of the programs that were synthesized for each drawing. Symmetric figures cluster to the right; "loopy" figures cluster to the left; complicated programs are at the top and simple programs are at the bottom.

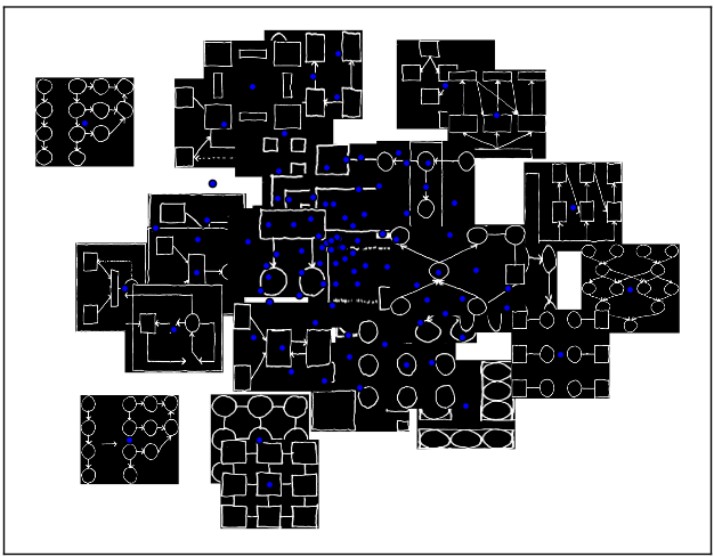

Figure 3: MDS on drawings using the learned distance metric, $L_{\text{learned}}(\cdot|\cdot)$. Drawings with similar looking parts in similar locations are clustered together.

## 4 NEURAL NETWORK ARCHITECTURE

### 4.1 HIGH-LEVEL OVERVIEW

For the model in Fig. 4, the distribution over the next drawing command factorizes as:

$$\mathbb{P}_\theta[t_1 t_2 \cdots t_K | I, T] = \prod_{k=1}^{K} \mathbb{P}_\theta \left[ t_k | a_\theta \left( f_\theta(I, \text{render}(T)) | \{t_j\}_{j=1}^{k-1} \right), \{t_j\}_{j=1}^{k-1} \right] \tag{7}$$

where $t_1 t_2 \cdots t_K$ are the tokens in the drawing command, $I$ is the target image, $T$ is a trace set, $\theta$ are the parameters of the neural network, $f_\theta(\cdot, \cdot)$ is the image feature extractor (convolutional network),

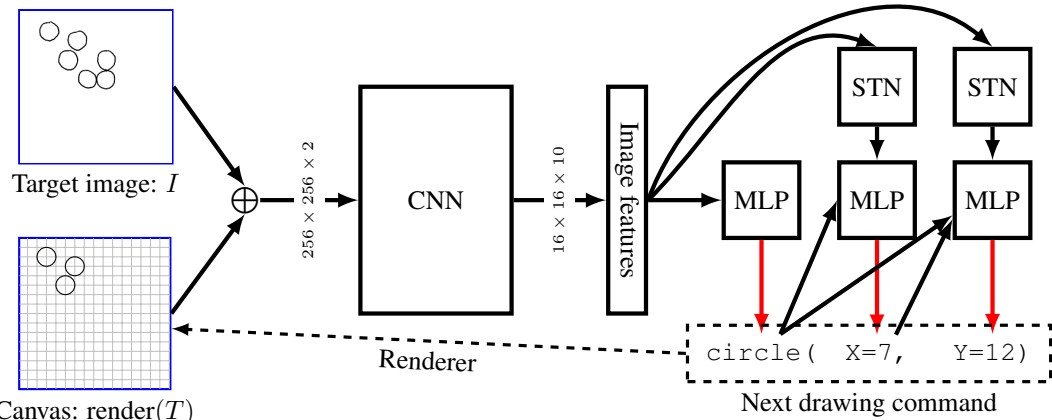

Figure 4: Our neural architecture for inferring the trace set of a graphics program from its output. Blue: network inputs. Black: network operations. Red: samples from a multinomial. Typewriter font: network outputs. Renders snapped to a $16 \times 16$ grid, illustrated in gray. STN (spatial transformer network) is a differentiable attention mechanism (2).

and $a_\theta(\cdot|\cdot)$ is an attention mechanism. The distribution over traces factorizes as:

$$\mathbb{P}_\theta[T|I] = \prod_{n=1}^{|T|} \mathbb{P}_\theta[T_n|I, T_{1:(n-1)}] \times \mathbb{P}_\theta[\texttt{STOP}|I, T] \tag{8}$$

where $|T|$ is the length of trace $T$, the subscripts on $T$ index drawing commands within the trace (so $T_n$ is a sequence of tokens: $t_1 t_2 \cdots t_K$), and the STOP token is emitted by the network to signal that the trace explains the image.

## 4.2 CONVOLUTIONAL NETWORK

The convolutional network takes as input 2 $256 \times 256$ images represented as a $2 \times 256 \times 256$ volume. These are passed through two layers of convolutions separated by ReLU nonlinearities and max pooling:

- Layer 1: 20 $8 \times 8$ convolutions, 2 $16 \times 4$ convolutions, 2 $4 \times 16$ convolutions. Followed by $8 \times 8$ pooling with a stride size of 4.
- Layer 2: 10 $8 \times 8$ convolutions. Followed by $4 \times 4$ pooling with a stride size of 4.

## 4.3 AUTOREGRESSIVE DECODING OF DRAWING COMMANDS

Given the image features $f$, we predict the first token (i.e., the name of the drawing command: circle, rectangle, line, or STOP) using logistic regression:

$$\mathbb{P}[t_1] \propto \exp\left(W_{t_1} f + b_{t_1}\right) \tag{9}$$

where $W_{t_1}$ is a learned weight matrix and $b_{t_1}$ is a learned bias vector.

Given an attention mechanism $a(\cdot|\cdot)$, subsequent tokens are predicted as:

$$\mathbb{P}[t_n|t_{1:(n-1)}] \propto \mathrm{MLP}_{t_1,n}(a(f|t_{1:(n-1)}) \oplus \bigoplus_{j<n} \mathrm{oneHot}(t_j)) \tag{10}$$

Thus each token of each drawing primitive has its own learned MLP. For predicting the coordinates of lines we found that using 32 hidden nodes with sigmoid activations worked well; for other tokens the MLP's are just logistic regression (no hidden nodes).

We use Spatial Transformer Networks (2) as our attention mechanism. The parameters of the spatial transform are predicted on the basis of previously predicted tokens. For example, in order to decide where to focus our attention when predicting the $y$ coordinate of a circle, we condition upon both the identity of the drawing command (circle) and upon the value of the previously predicted $x$ coordinate:

$$a(f|t_{1:(n-1)}) = \text{AffineTransform}(f, \text{MLP}_{t_1,n}(\bigoplus_{j<n} \text{oneHot}(t_j))) \tag{11}$$

So, we learn a different network for predicting special transforms *for each drawing command* (value of $t_1$) and also *for each token of the drawing command*. These networks ($\text{MLP}_{t_1,n}$ in equation 11) have no hidden layers and output the 6 entries of an affine transformation matrix; see (2) for more details.

Training takes a little bit less than a day on a Nvidia TitanX GPU. The network was trained on $10^5$ synthetic examples.

## 4.4 LSTM BASELINE

We compared our deep network with a baseline that models the problem as a kind of image captioning. Given the target image, this baseline produces the program trace in one shot by using a CNN to extract features of the input which are passed to an LSTM which finally predicts the trace token-by-token. This general architecture is used in several successful neural models of image captioning (e.g., (4)).

Concretely, we kept the image feature extractor architecture (a CNN) as in our model, but only passed it one image as input (the target image to explain). Then, instead of using an autoregressive decoder to predict a single drawing command, we used an LSTM to predict a sequence of drawing commands token-by-token. This LSTM had 128 memory cells, and at each time step produced as output the next token in the sequence of drawing commands. It took as input both the image representation and its previously predicted token.

## 4.5 A LEARNED LIKELIHOOD SURROGATE

Our architecture for $L_{\text{learned}}(\text{render}(T_1)|\text{render}(T_2))$ has the same series of convolutions as the network that predicts the next drawing command. We train it to predict two scalars: $|T_1 - T_2|$ and $|T_2 - T_1|$. These predictions are made using linear regression from the image features followed by a ReLU nonlinearity; this nonlinearity makes sense because the predictions can never be negative but could be arbitrarily large positive numbers.

We train this network by sampling random synthetic scenes for $T_1$, and then perturbing them in small ways to produce $T_2$. We minimize the squared loss between the network's prediction and the ground truth symmetric differences. $T_1$ is rendered in a "simulated hand drawing" style which we describe next.

## 5 SIMULATING HAND DRAWINGS

We introduce noise into the LaTeX rendering process by:

- Rescaling the image intensity by a factor chosen uniformly at random from $[0.5, 1.5]$
- Translating the image by $\pm 3$ pixels chosen uniformly random
- Rendering the LaTeX using the pencildraw style, which adds random perturbations to the paths drawn by LaTeX in a way designed to resemble a pencil.
- Randomly perturbing the positions and sizes of primitive LaTeX drawing commands

## 6 LIKELIHOOD SURROGATE FOR SYNTHETIC DATA

For synthetic data (e.g., LaTeX output) it is relatively straightforward to engineer an adequate distance measure between images, because it is possible for the system to discover drawing commands that

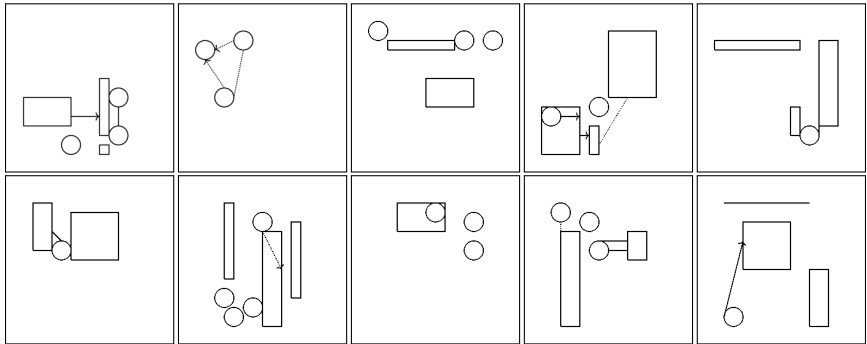

Figure 5: Example synthetic training data

exactly match the pixels in the target image. We use:

$$-\log L(I_1|I_2) = \sum_{1 \le x \le 256} \sum_{1 \le y \le 256} |I_1[x,y] - I_2[x,y]| \begin{cases} \alpha, \text{ if } I_1[x,y] > I_2[x,y] \\ \beta, \text{ if } I_1[x,y] < I_2[x,y] \\ 0, \text{ if } I_1[x,y] = I_2[x,y] \end{cases} \tag{12}$$

where $\alpha$, $\beta$ are constants that control the trade-off between preferring to explain the pixels in the image (at the expense of having extraneous pixels) and not predicting pixels where they don't exist (at the expense of leaving some pixels unexplained). Because our sampling procedure incrementally constructs the scene part-by-part, we want $\alpha > \beta$. That is, it is preferable to leave some pixels unexplained; for once a particle in SMC adds a drawing primitive to its trace that is not actually in the latent scene, it can never recover from this error. In our experiments on synthetic data we used $\alpha = 0.8$ and $\beta = 0.04$.

## 7 GENERATING SYNTHETIC TRAINING DATA

We generated synthetic training data for the neural network by sampling LaTeX code according to the following generative process: First, the number of objects in the scene are sampled uniformly from 1 to 12. For each object we uniformly sample its identity (circle, rectangle, or line). Then we sample the parameters of the circles, than the parameters of the rectangles, and finally the parameters of the lines; this has the effect of teaching the network to first draw the circles in the scene, then the rectangles, and finally the lines. We furthermore put the circle (respectively, rectangle and line) drawing commands in order by left-to-right, bottom-to-top; thus the training data enforces a canonical order in which to draw any scene.

To make the training data look more like naturally occurring figures, we put a Chinese restaurant process prior (5) over the values of the X and Y coordinates that occur in the execution trace. This encourages reuse of coordinate values, and so produces training data that tends to have parts that are nicely aligned.

In the synthetic training data we excluded any sampled scenes that had overlapping drawing commands. As shown in the main paper, the network is then able to generalize to scenes with, for example, intersecting lines or lines that penetrate a rectangle.

When sampling the endpoints of a line, we biased the sampling process so that it would be more likely to start an endpoint along one of the sides of a rectangle or at the boundary of a circle. If $n$ is the number of points either along the side of a rectangle or at the boundary of a circle, we would sample an arbitrary endpoint with probability $\frac{2}{2+n}$ and sample one of the "attaching" endpoints with probability $\frac{1}{2+n}$.

See figure 7 for examples of the kinds of scenes that the network is trained on.

For readers wishing to generate their own synthetic training sets, we refer them to our source code at: `redactedForAnonymity.com`.

# 8 THE COST FUNCTION FOR PROGRAMS

We seek the minimum cost program which evaluates to (produces the drawing primitives in) an execution trace $T$:

$$\text{program}(T) = \underset{\substack{p \in \text{DSL} \\ p \text{ evaluates to } T}}{\arg \min} \text{cost}(p) \qquad (13)$$

Programs incur a cost of 1 for each command (primitive drawing action, loop, or reflection). They incur a cost of $\frac{1}{3}$ for each unique coefficient they use in a linear transformation beyond the first coefficient. This encourages reuse of coefficients, which leads to code that has translational symmetry; rather than provide a translational symmetry operator as we did with reflection, we modify what is effectively a prior over the space of program so that it tends to produce programs that have this symmetry.

Programs also incur a cost of 1 for having loops of constant length 2; otherwise there is often no pressure from the cost function to explain a repetition of length 2 as being a reflection rather a loop.

# 9 FULL RESULTS ON DRAWINGS DATA SET

Below we show our full data set of drawings. The leftmost column is a hand drawing. The middle column is a rendering of the most likely trace discovered by the neurally guided SMC sampling scheme. The rightmost column is the program we synthesized from a ground truth execution trace of the drawing. Note that because the inference procedure is stochastic, the top one most likely sample can vary from run to run. Below we report a representative sample from a run with 2000 particles.

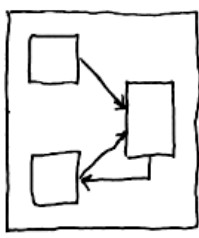
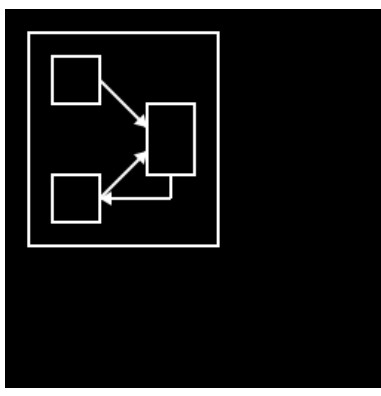

```
line(6,2,6,3,
arrow = False,solid = True);
line(6,2,3,2,
arrow = True,solid = True);
reflect(y = 9){
line(3,7,5,5,
arrow = True,solid = True);
rectangle(1,1,3,3);
rectangle(5,3,7,6);
rectangle(0,0,8,9)
}
```

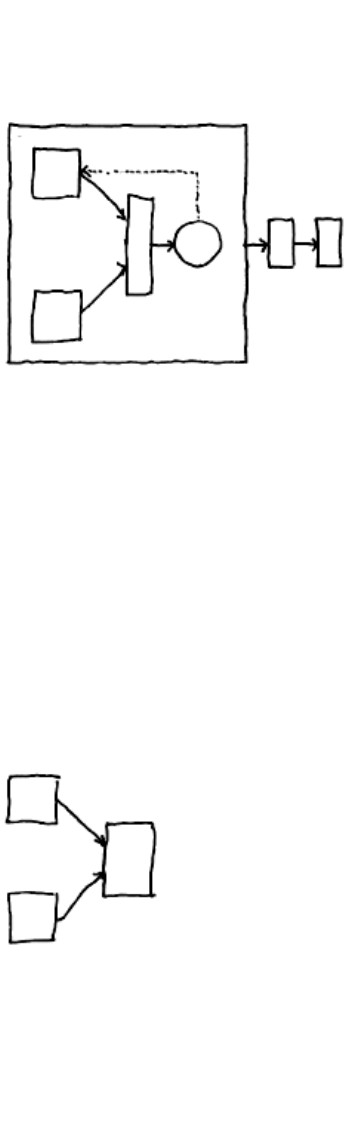

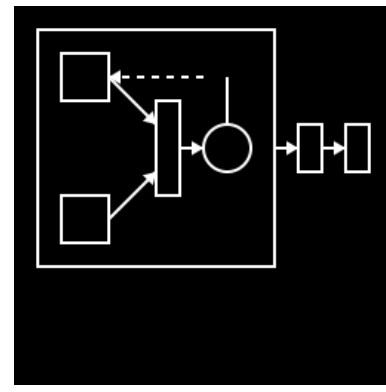

```
for (i < 2){
line(8,8,3,8,
arrow = True,solid = False);
line(-2 * i + 12,5,-2 * i + 13,5
arrow = True,solid = True);
line(6,5,7,5,
arrow = True,solid = True);
line(3,-6 * i + 8,5,-2 * i + 6,
arrow = True,solid = True);
rectangle(-2 * i + 13,4,-2 * i +
rectangle(1,-6 * i + 7,3,-6 * i
};
circle(8,5);
rectangle(5,3,6,7);
rectangle(0,0,10,10);
line(8,6,8,8,
arrow = False,solid = False)
```

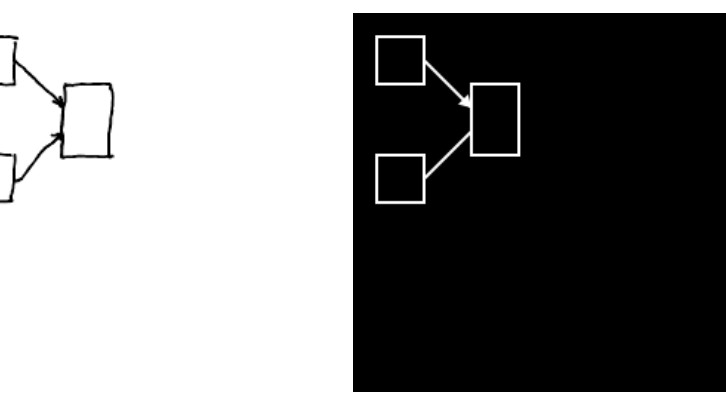

```
reflect(y = 7){
line(2,6,4,4,
arrow = True,solid = True);
rectangle(0,0,2,2)
};
rectangle(4,2,6,5)
```

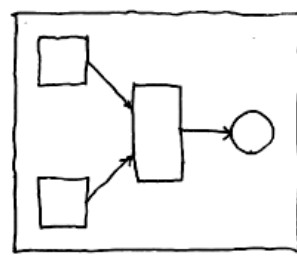

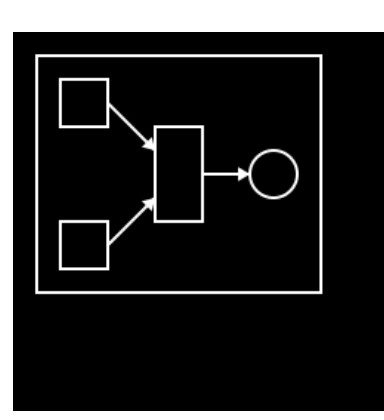

```
line(7,5,9,5,
arrow = True,solid = True);
rectangle(5,3,7,7);
rectangle(0,0,12,10);
reflect(y = 10){
circle(10,5);
line(3,2,5,4,
arrow = True,solid = True);
rectangle(1,1,3,3)
}
```

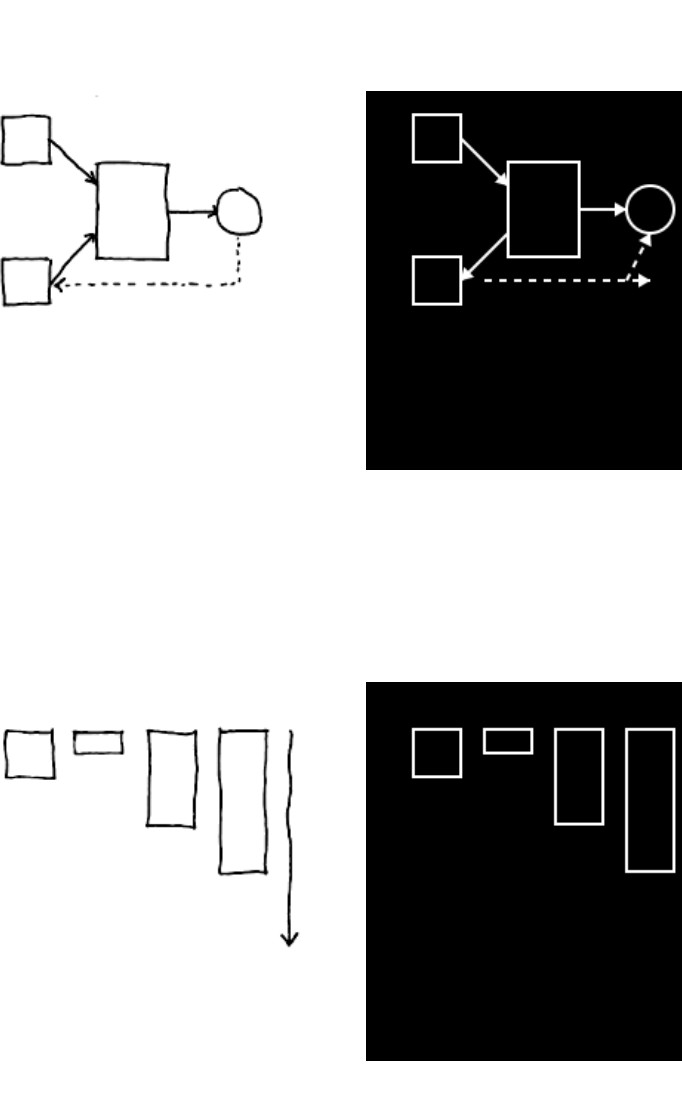

```
line(10,1,2,1,
arrow = True,solid = False);
line(10,1,10,3,
arrow = False,solid = False);
line(7,4,9,4,
arrow = True,solid = True);
reflect(y = 8){
circle(10,4);
line(2,1,4,3,
arrow = True,solid = True);
rectangle(4,2,7,6);
rectangle(0,6,2,8)
}
```

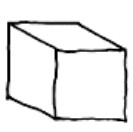

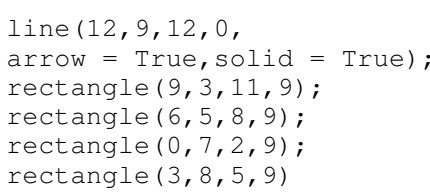

```
line(12,9,12,0,
arrow = True,solid = True);
rectangle(9,3,11,9);
rectangle(6,5,8,9);
rectangle(0,7,2,9);
rectangle(3,8,5,9)
```

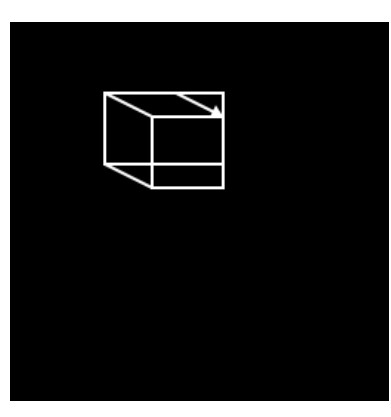

```
for (i < 3){
for (j < (1*i + 1)){
if (j > 0){
line(3 * j + −3,3 * i + −2,3 * j
arrow = False,solid = True);
line(0,3 * j + −2,3 * j + −3,4,
arrow = False,solid = True)
}
rectangle(2,0,5,3)
}
}
```

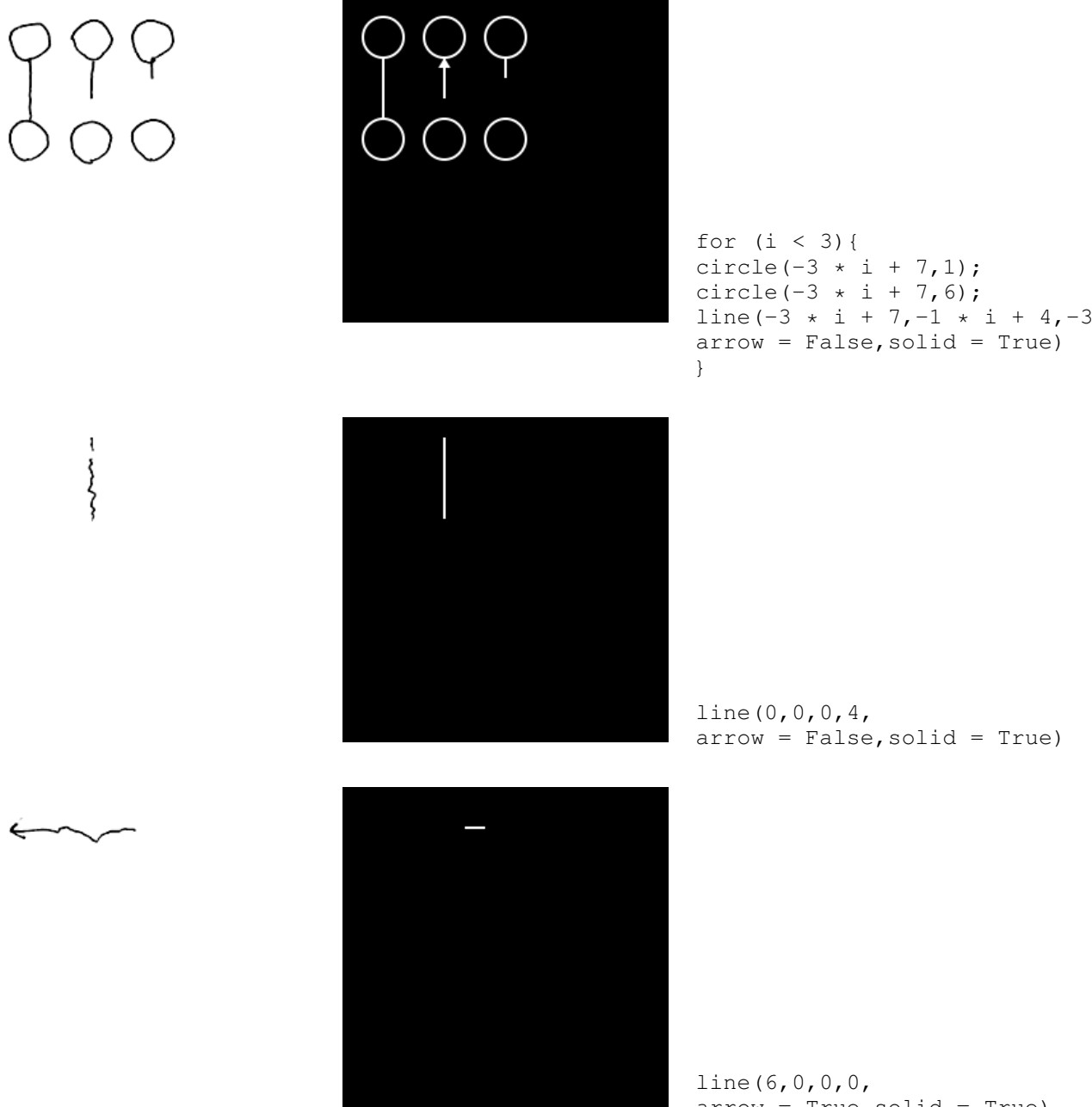

```
for (i < 3){
circle(-3 * i + 7,1);
circle(-3 * i + 7,6);
line(-3 * i + 7,-1 * i + 4,-3 *
arrow = False,solid = True)
}
```

```
line(0,0,0,4,
arrow = False,solid = True)
```

```
line(6,0,0,0,
arrow = True,solid = True)
```

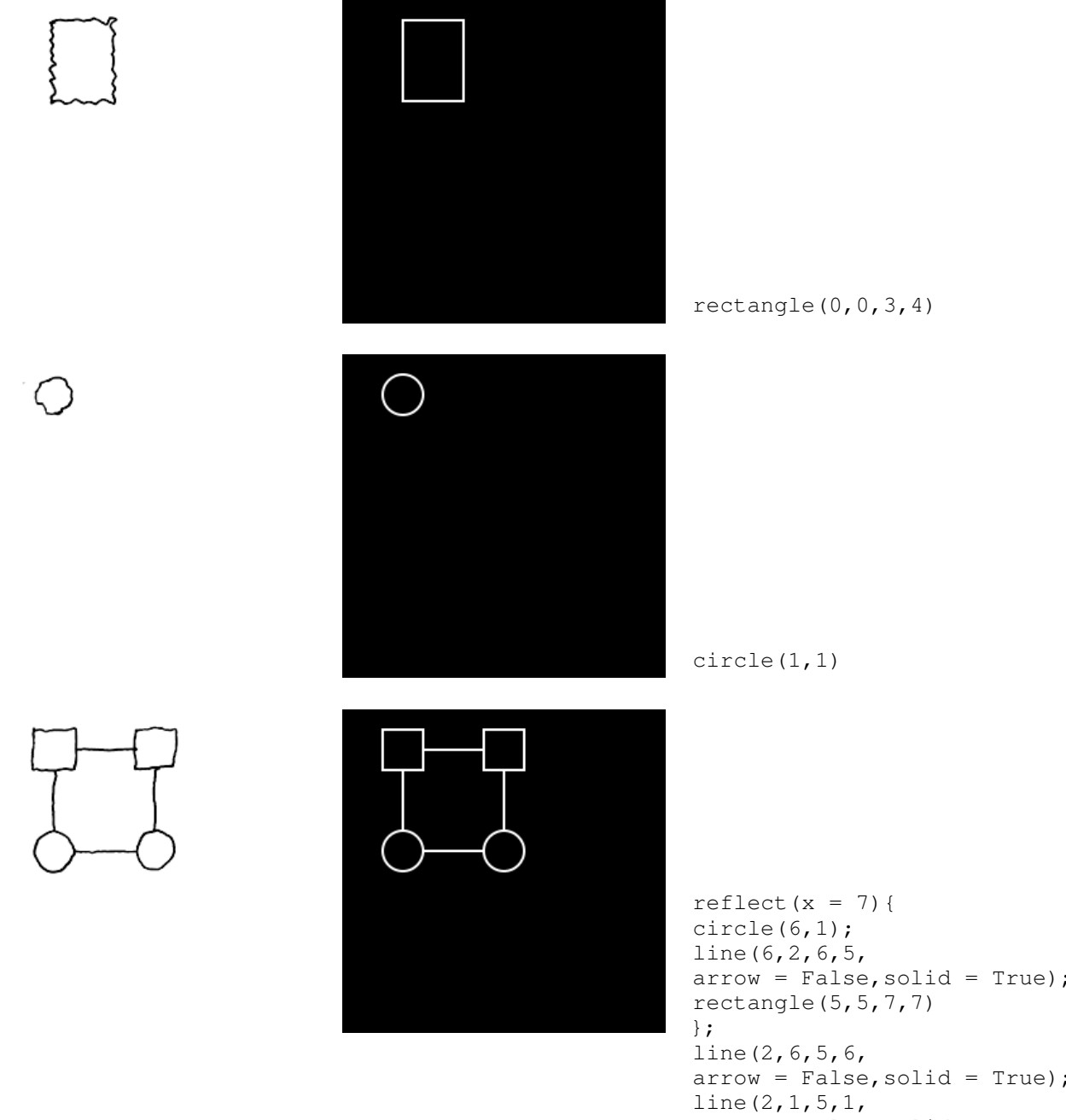

```
rectangle(0,0,3,4)
```

```
circle(1,1)
```

```
reflect(x = 7){
circle(6,1);
line(6,2,6,5,
arrow = False,solid = True);
rectangle(5,5,7,7)
};
line(2,6,5,6,
arrow = False,solid = True);
line(2,1,5,1,
arrow = False,solid = True)
```

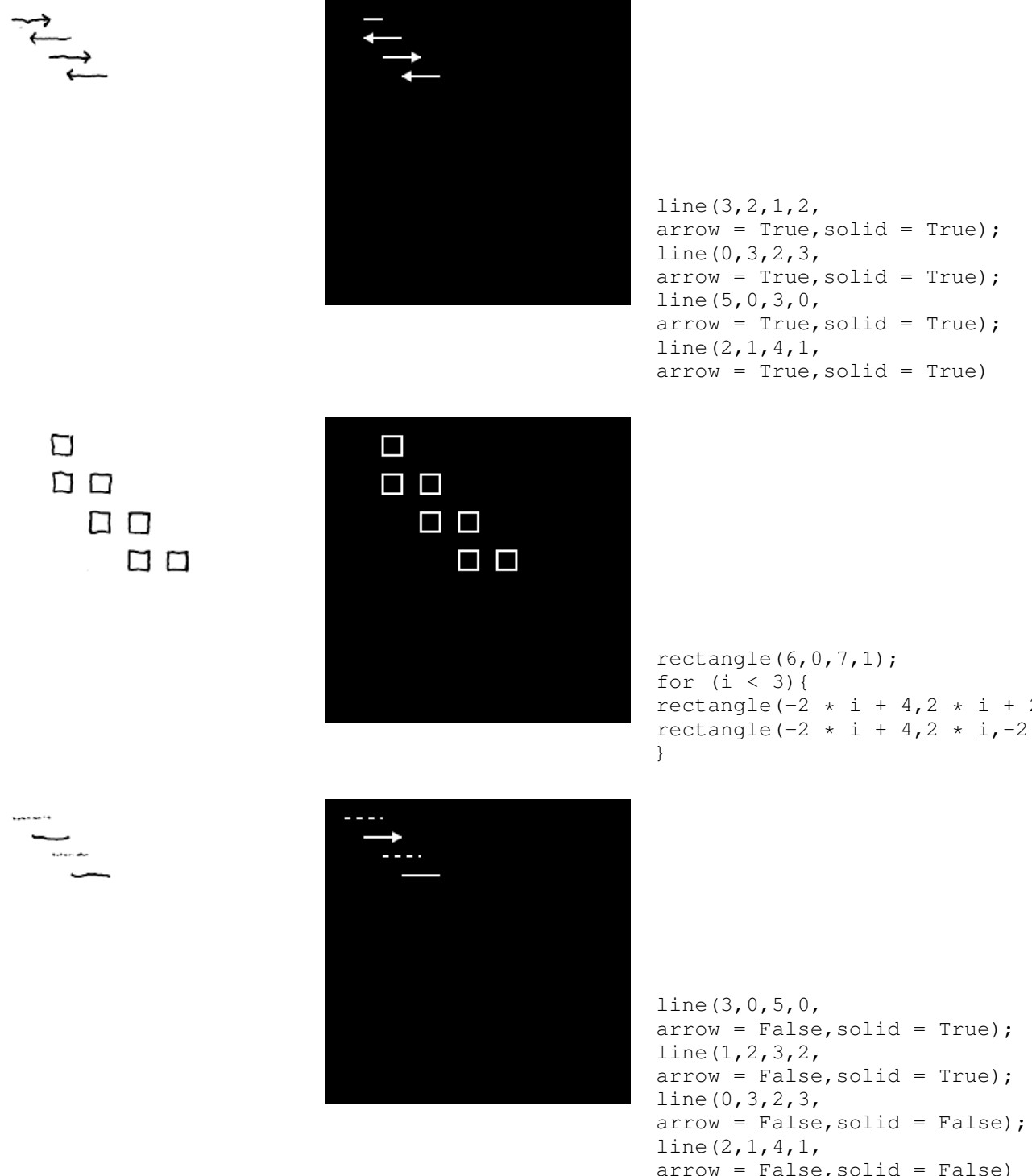

```
line(3,2,1,2,
arrow = True,solid = True);
line(0,3,2,3,
arrow = True,solid = True);
line(5,0,3,0,
arrow = True,solid = True);
line(2,1,4,1,
arrow = True,solid = True)
```

```
rectangle(6,0,7,1);
for (i < 3){
rectangle(-2 * i + 4,2 * i + 2,-
rectangle(-2 * i + 4,2 * i,-2 *
}
```

```
line(3,0,5,0,
arrow = False,solid = True);
line(1,2,3,2,
arrow = False,solid = True);
line(0,3,2,3,
arrow = False,solid = False);
line(2,1,4,1,
arrow = False,solid = False)
```

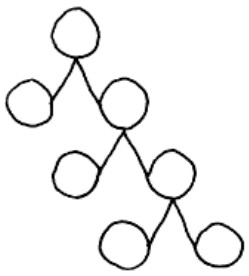 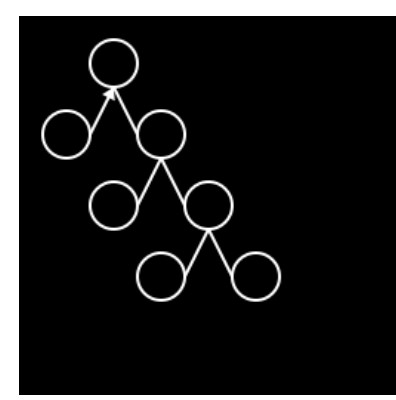

```
circle(9,1);
for (i < 3){
circle(-2 * i + 7,3 * i + 4);
circle(-2 * i + 5,3 * i + 1);
line(-2 * i + 6,3 * i + 1,-2 * i
arrow = False,solid = True);
line(-2 * i + 7,3 * i + 3,-2 * i
arrow = False,solid = True)
}
```

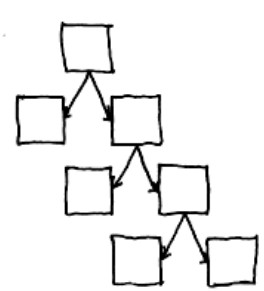 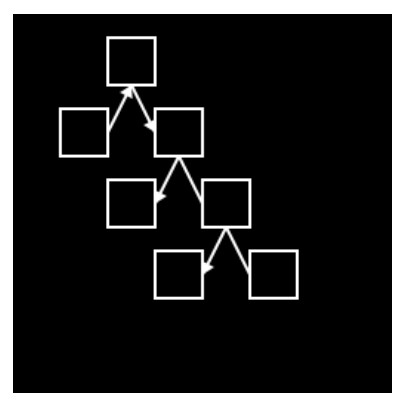

```
for (i < 3){
line(2 * i + 3,-3 * i + 9,2 * i
arrow = True,solid = True);
line(2 * i + 3,-3 * i + 9,2 * i
arrow = True,solid = True);
rectangle(2 * i + 2,-3 * i + 9,2
rectangle(2 * i,-3 * i + 6,2 * i
};
rectangle(8,0,10,2)
```

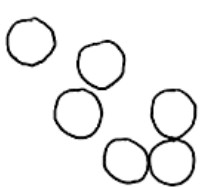 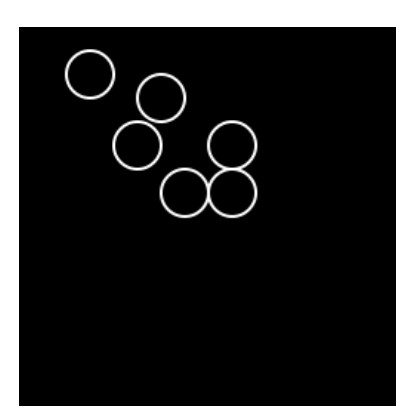

```
for (i < 2){
circle(2 * i + 1,-3 * i + 6);
circle(-3 * i + 7,2 * i + 3);
circle(2 * i + 5,1)
}
```

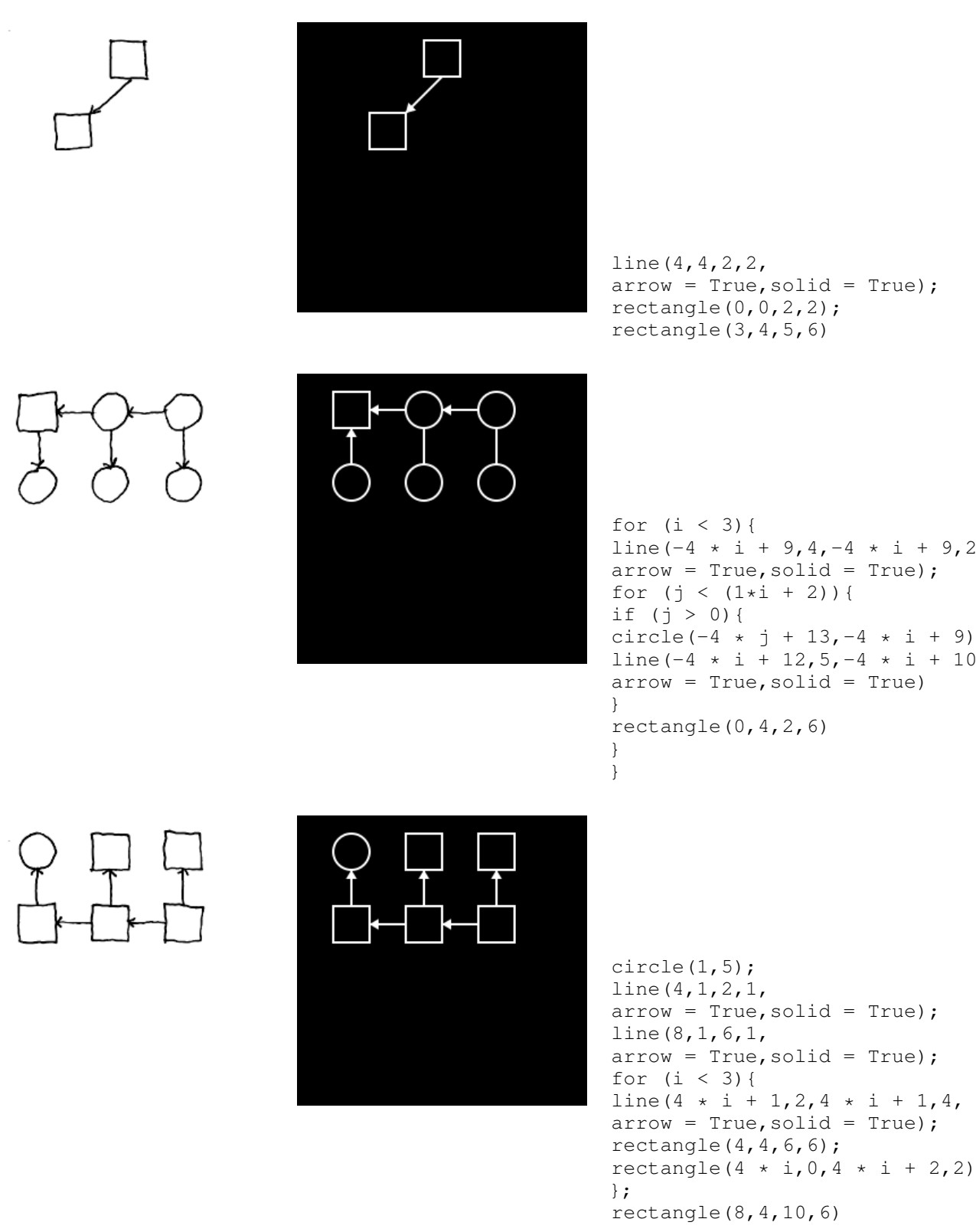

```
line(4,4,2,2,
arrow = True,solid = True);
rectangle(0,0,2,2);
rectangle(3,4,5,6)
```

```
for (i < 3){
line(-4 * i + 9,4,-4 * i + 9,2,
arrow = True,solid = True);
for (j < (1*i + 2)){
if (j > 0){
circle(-4 * j + 13,-4 * i + 9);
line(-4 * i + 12,5,-4 * i + 10,5
arrow = True,solid = True)
}
rectangle(0,4,2,6)
}
}
```

```
circle(1,5);
line(4,1,2,1,
arrow = True,solid = True);
line(8,1,6,1,
arrow = True,solid = True);
for (i < 3){
line(4 * i + 1,2,4 * i + 1,4,
arrow = True,solid = True);
rectangle(4,4,6,6);
rectangle(4 * i,0,4 * i + 2,2)
};
rectangle(8,4,10,6)
```

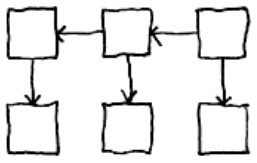

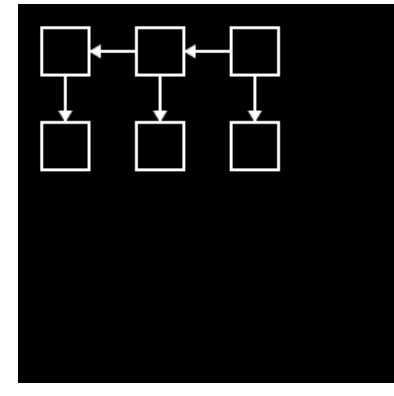

```
for (i < 3){
line(4 * i + 1,4,4 * i + 1,2,
arrow = True,solid = True);
for (j < 2){
line(4 * j + 4,5,4 * j + 2,5,
arrow = True,solid = True);
rectangle(4 * i,-4 * j + 4,4 * i
}
}
```

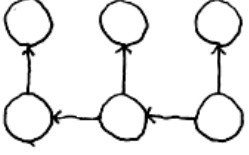

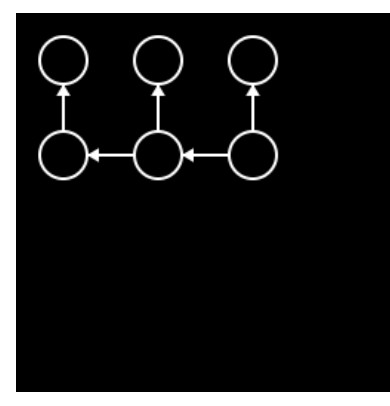

```
for (i < 3){
line(4 * i + 1,2,4 * i + 1,4,
arrow = True,solid = True);
for (j < 2){
circle(4 * i + 1,4 * j + 1);
line(4 * j + 4,1,4 * j + 2,1,
arrow = True,solid = True)
}
}
```

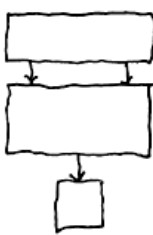

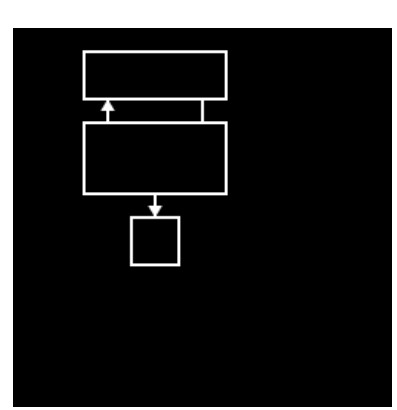

```
line(5,7,5,6,
arrow = True,solid = True);
line(3,3,3,2,
arrow = True,solid = True);
line(1,7,1,6,
arrow = True,solid = True);
rectangle(0,3,6,6);
rectangle(2,0,4,2);
rectangle(0,7,6,9)
```

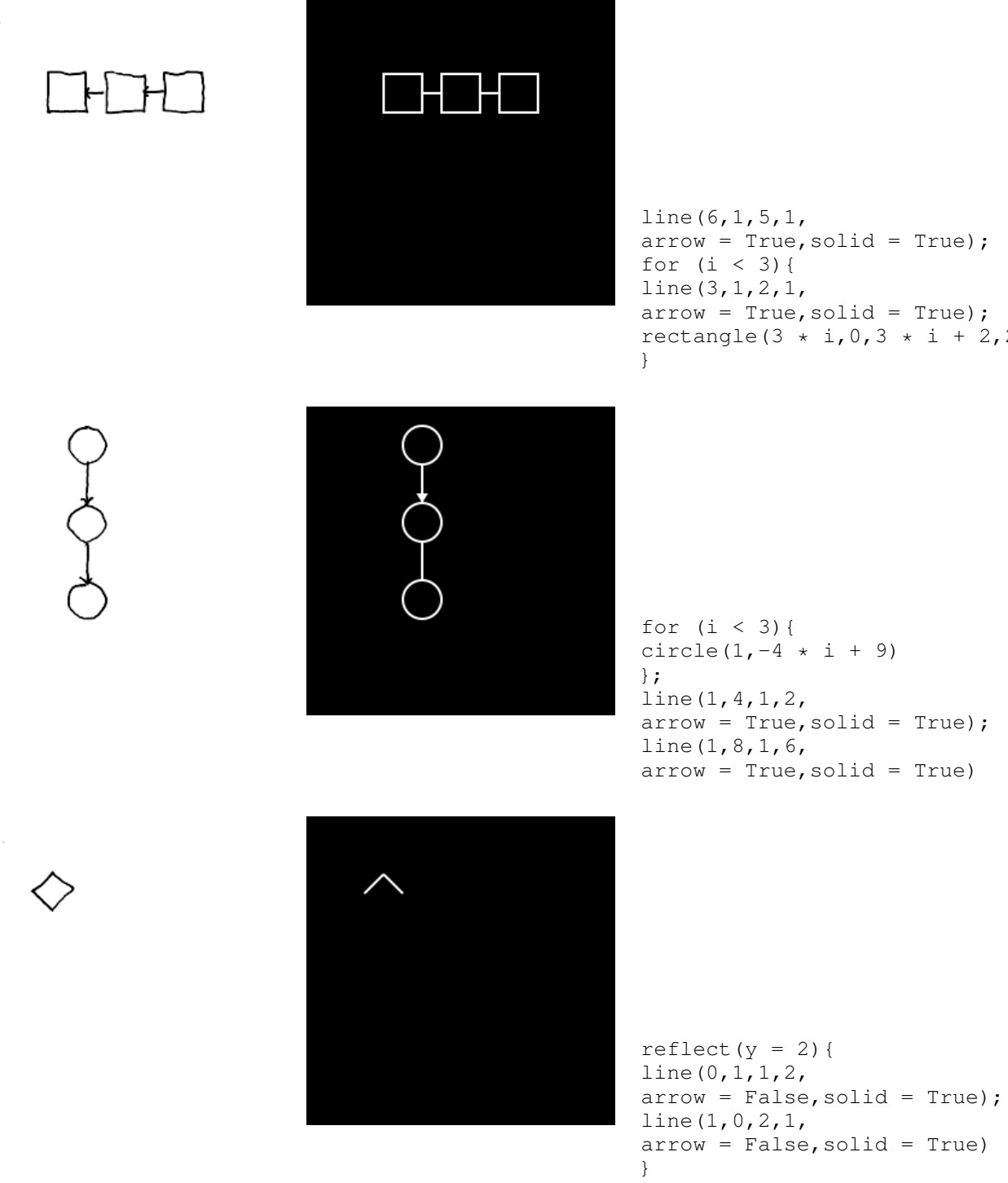

```
line(6,1,5,1,
arrow = True,solid = True);
for (i < 3){
line(3,1,2,1,
arrow = True,solid = True);
rectangle(3 * i,0,3 * i + 2,2)
}
```

```
for (i < 3){
circle(1,-4 * i + 9)
};
line(1,4,1,2,
arrow = True,solid = True);
line(1,8,1,6,
arrow = True,solid = True)
```

```
reflect(y = 2){
line(0,1,1,2,
arrow = False,solid = True);
line(1,0,2,1,
arrow = False,solid = True)
}
```

```
line(0,0,0,2,
arrow = False,solid = True);
line(0,2,2,2,
arrow = False,solid = True)
```

```
for (i < 3){
line(1 * i,-2 * i + 4,1 * i,-1 *
arrow = False,solid = True);
line(1 * i,-1 * i + 6,2 * i + 2,
arrow = False,solid = True)
}
```

```
circle(5,2);
circle(5,4);
rectangle(4,1,6,5);
reflect(y = 5){
circle(1,4);
rectangle(0,3,2,5)
}
```

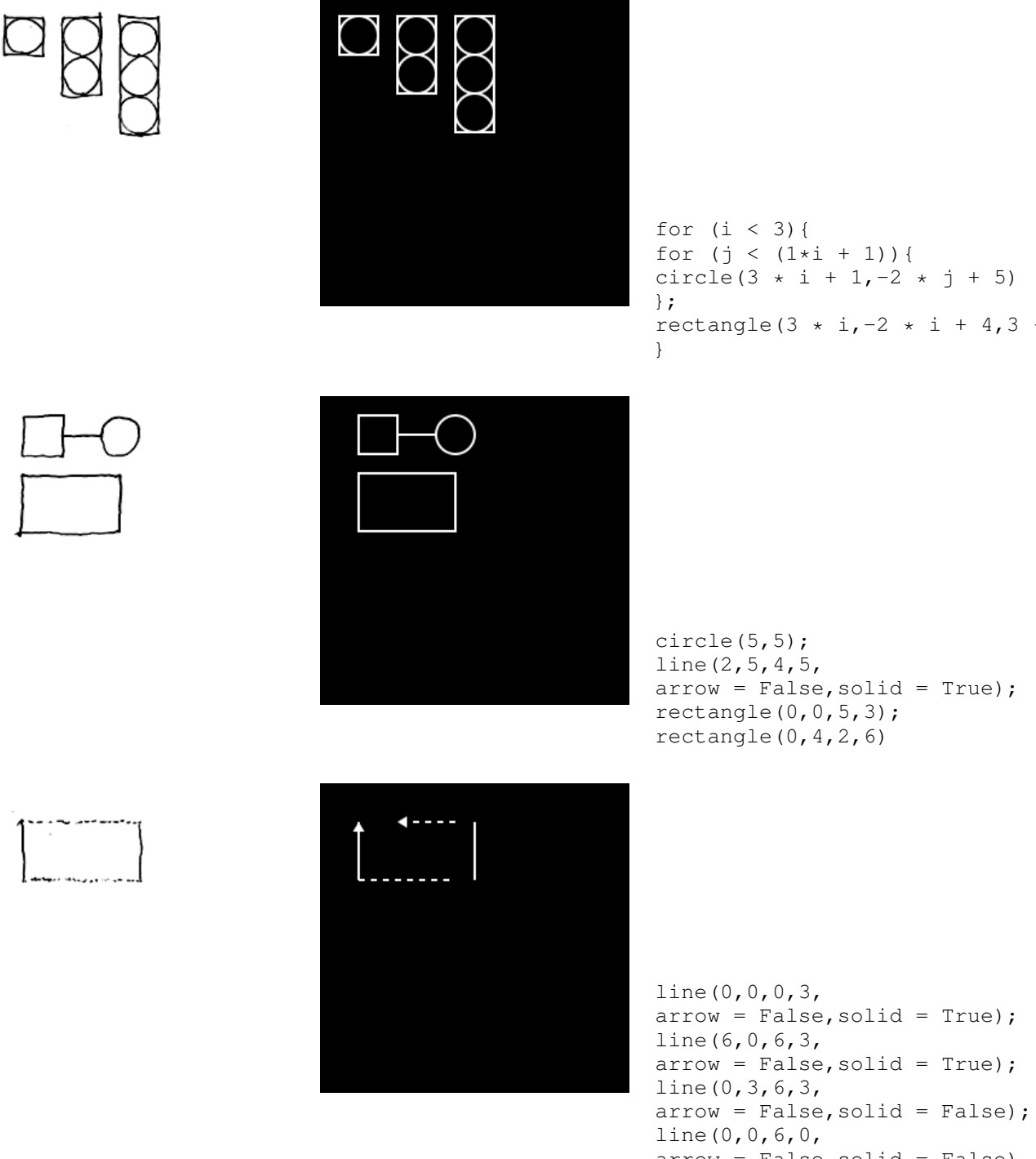

```
for (i < 3){
for (j < (1*i + 1)){
circle(3 * i + 1,-2 * j + 5)
};
rectangle(3 * i,-2 * i + 4,3 * i
}
```

```
circle(5,5);
line(2,5,4,5,
arrow = False,solid = True);
rectangle(0,0,5,3);
rectangle(0,4,2,6)
```

```
line(0,0,0,3,
arrow = False,solid = True);
line(6,0,6,3,
arrow = False,solid = True);
line(0,3,6,3,
arrow = False,solid = False);
line(0,0,6,0,
arrow = False,solid = False)
```

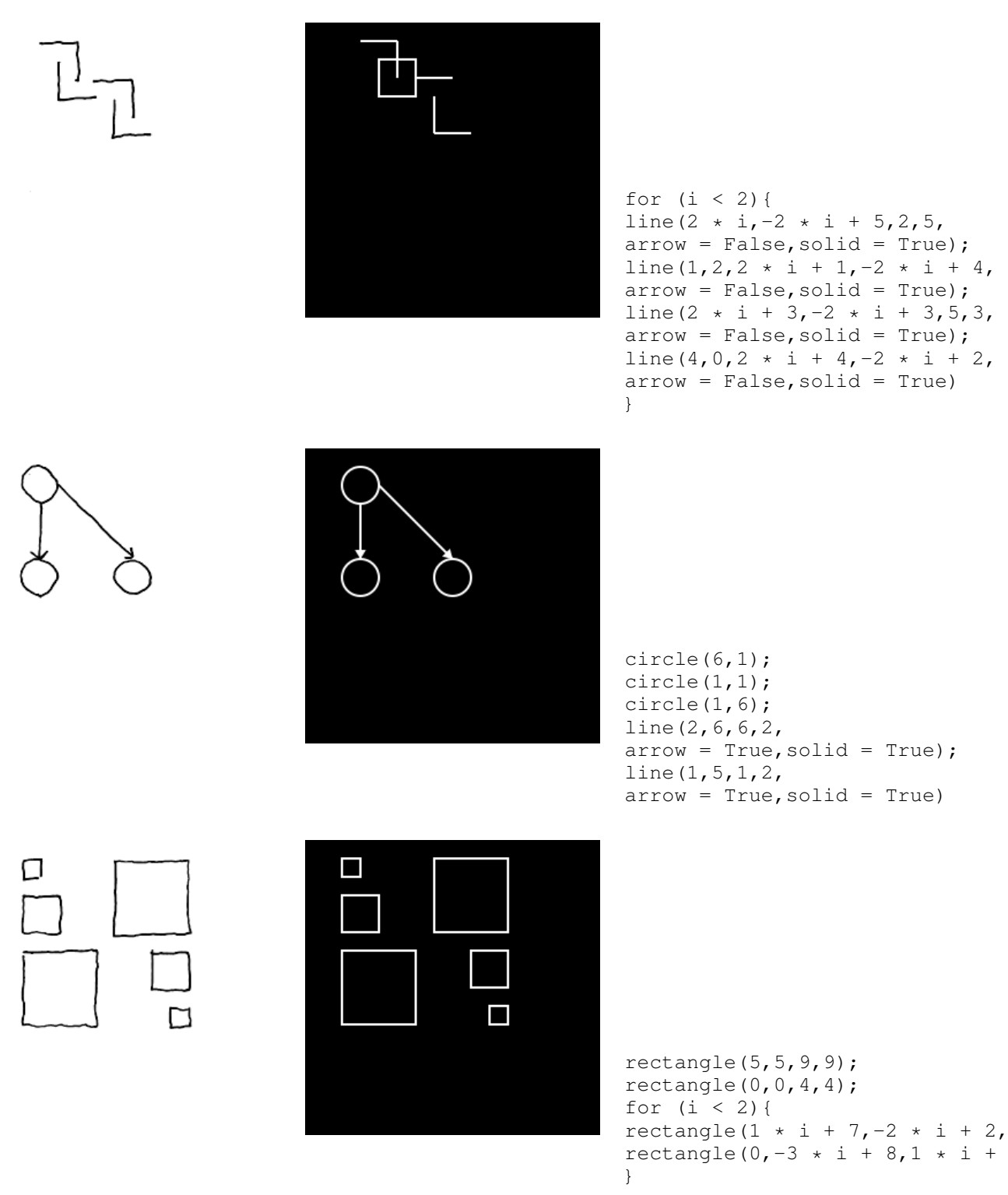

```
for (i < 2){
line(2 * i,-2 * i + 5,2,5,
arrow = False,solid = True);
line(1,2,2 * i + 1,-2 * i + 4,
arrow = False,solid = True);
line(2 * i + 3,-2 * i + 3,5,3,
arrow = False,solid = True);
line(4,0,2 * i + 4,-2 * i + 2,
arrow = False,solid = True)
}
```

```
circle(6,1);
circle(1,1);
circle(1,6);
line(2,6,6,2,
arrow = True,solid = True);
line(1,5,1,2,
arrow = True,solid = True)
```

```
rectangle(5,5,9,9);
rectangle(0,0,4,4);
for (i < 2){
rectangle(1 * i + 7,-2 * i + 2,9
rectangle(0,-3 * i + 8,1 * i + 1
}
```

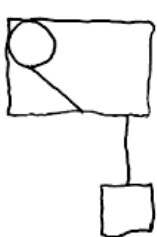

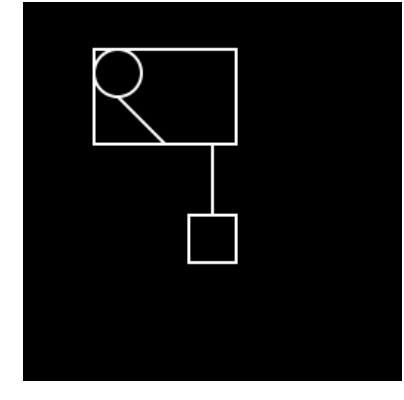

```
circle(1,8);
line(5,2,5,5,
arrow = False,solid = True);
line(1,7,3,5,
arrow = False,solid = True);
rectangle(4,0,6,2);
rectangle(0,5,6,9)
```

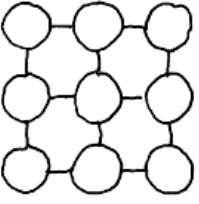

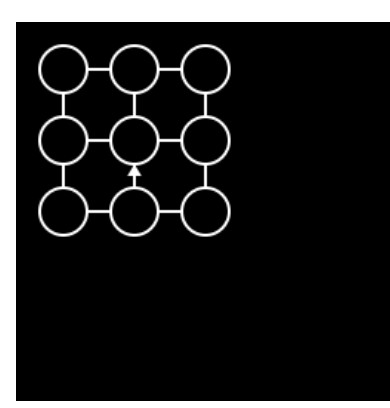

```
for (i < 3){
for (j < 3){
if (j > 0){
line(3 * j + -1,3 * i + 1,3 * j,
arrow = False,solid = True);
line(3 * i + 1,3 * j + -1,3 * i
arrow = False,solid = True)
}
circle(3 * i + 1,3 * j + 1)
}
}
```

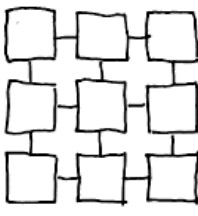

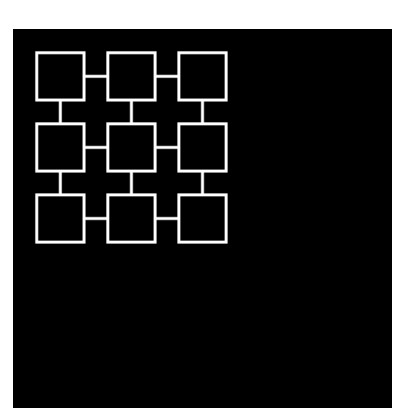

```
for (i < 3){
for (j < 3){
if (j > 0){
line(3 * i + 1,3 * j + -1,3 * i
arrow = False,solid = True);
line(3 * j + -1,3 * i + 1,3 * j,
arrow = False,solid = True)
}
rectangle(3 * i,3 * j,3 * i + 2,
}
}
```

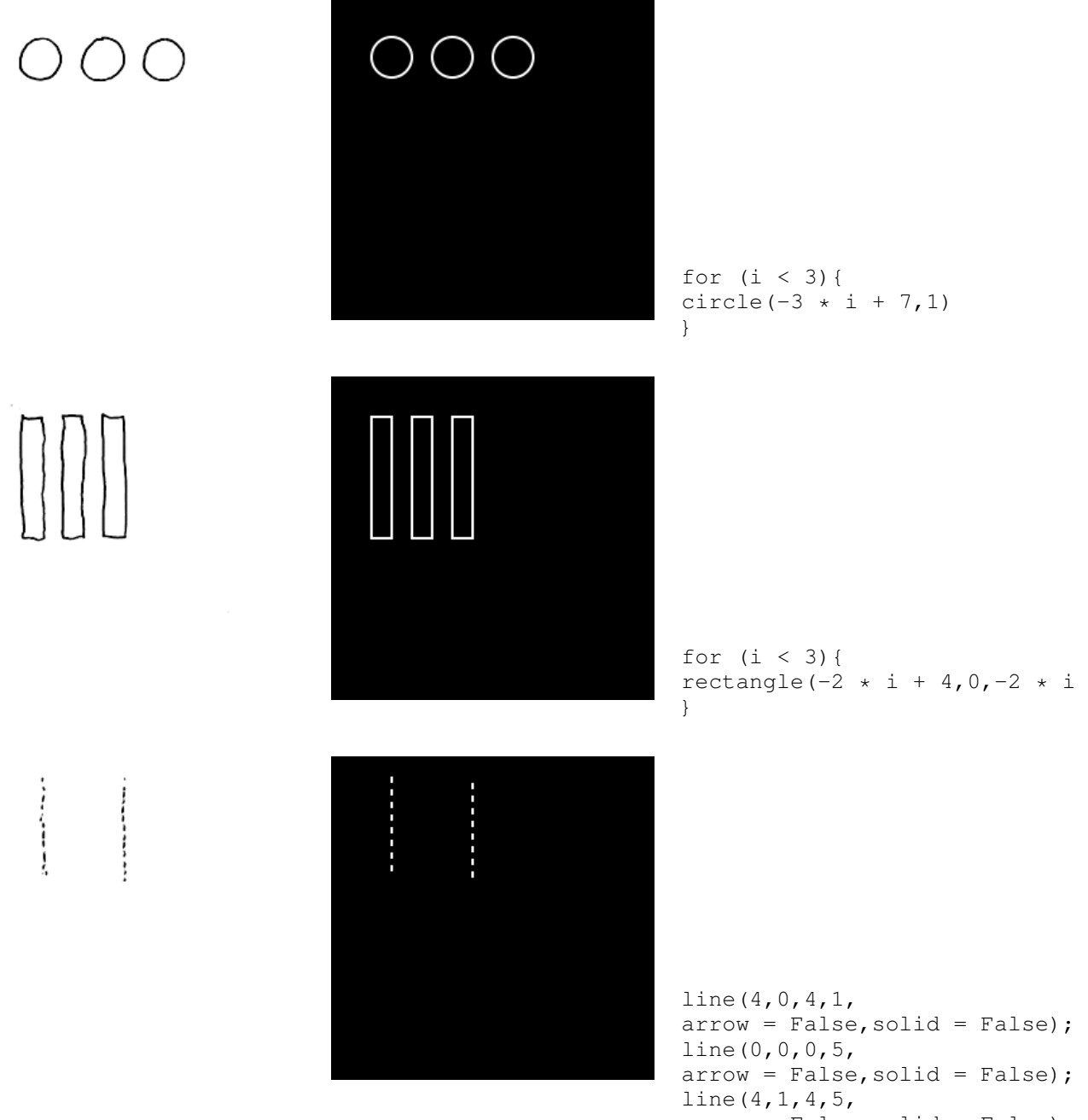

```
for (i < 3){
circle(-3 * i + 7,1)
}
```

```
for (i < 3){
rectangle(-2 * i + 4,0,-2 * i +
}
```

```
line(4,0,4,1,
arrow = False,solid = False);
line(0,0,0,5,
arrow = False,solid = False);
line(4,1,4,5,
arrow = False,solid = False)
```

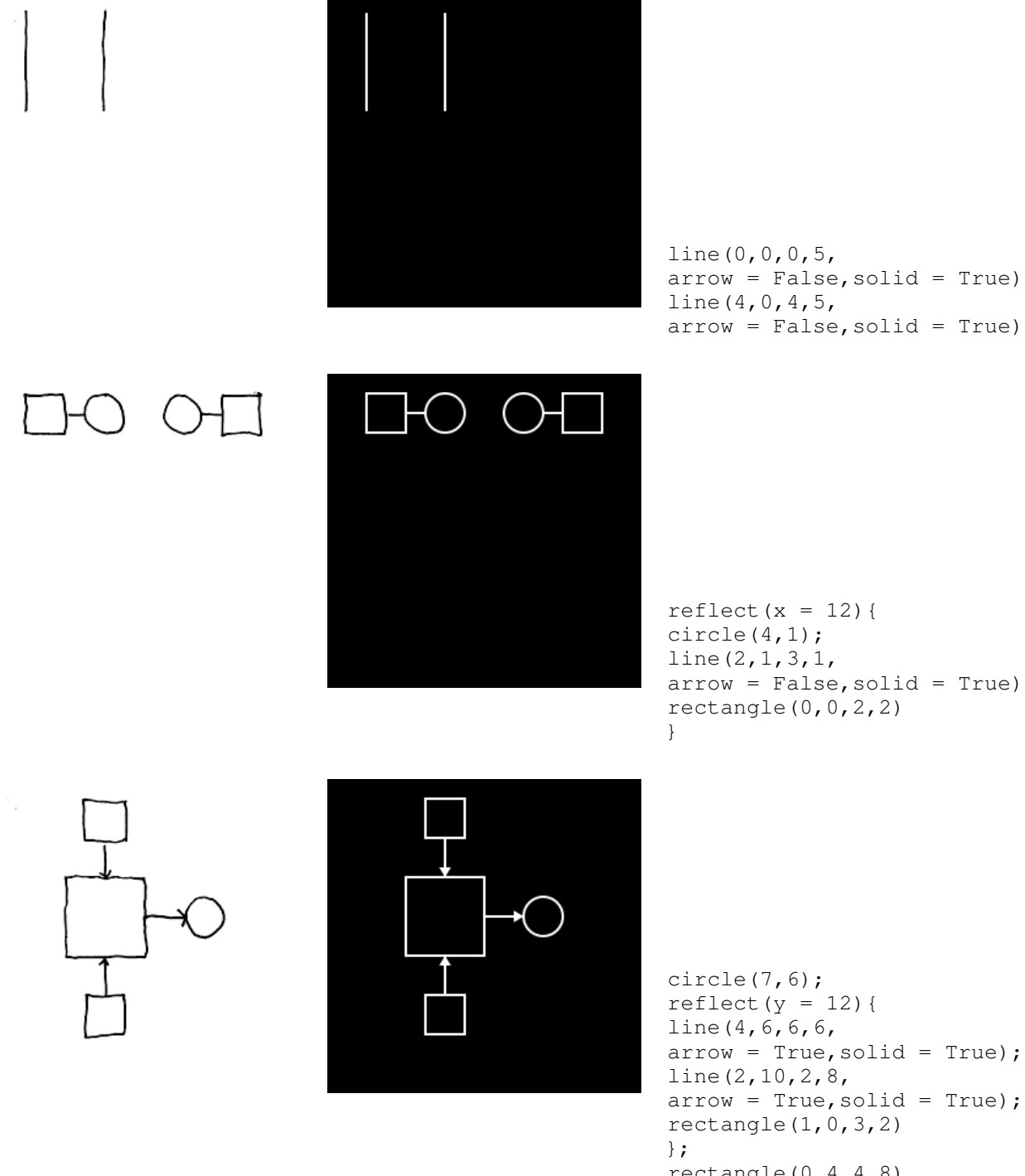

```
line(0,0,0,5,
arrow = False,solid = True);
line(4,0,4,5,
arrow = False,solid = True)
```

```
reflect(x = 12){
circle(4,1);
line(2,1,3,1,
arrow = False,solid = True);
rectangle(0,0,2,2)
}
```

```
circle(7,6);
reflect(y = 12){
line(4,6,6,6,
arrow = True,solid = True);
line(2,10,2,8,
arrow = True,solid = True);
rectangle(1,0,3,2)
};
rectangle(0,4,4,8)
```

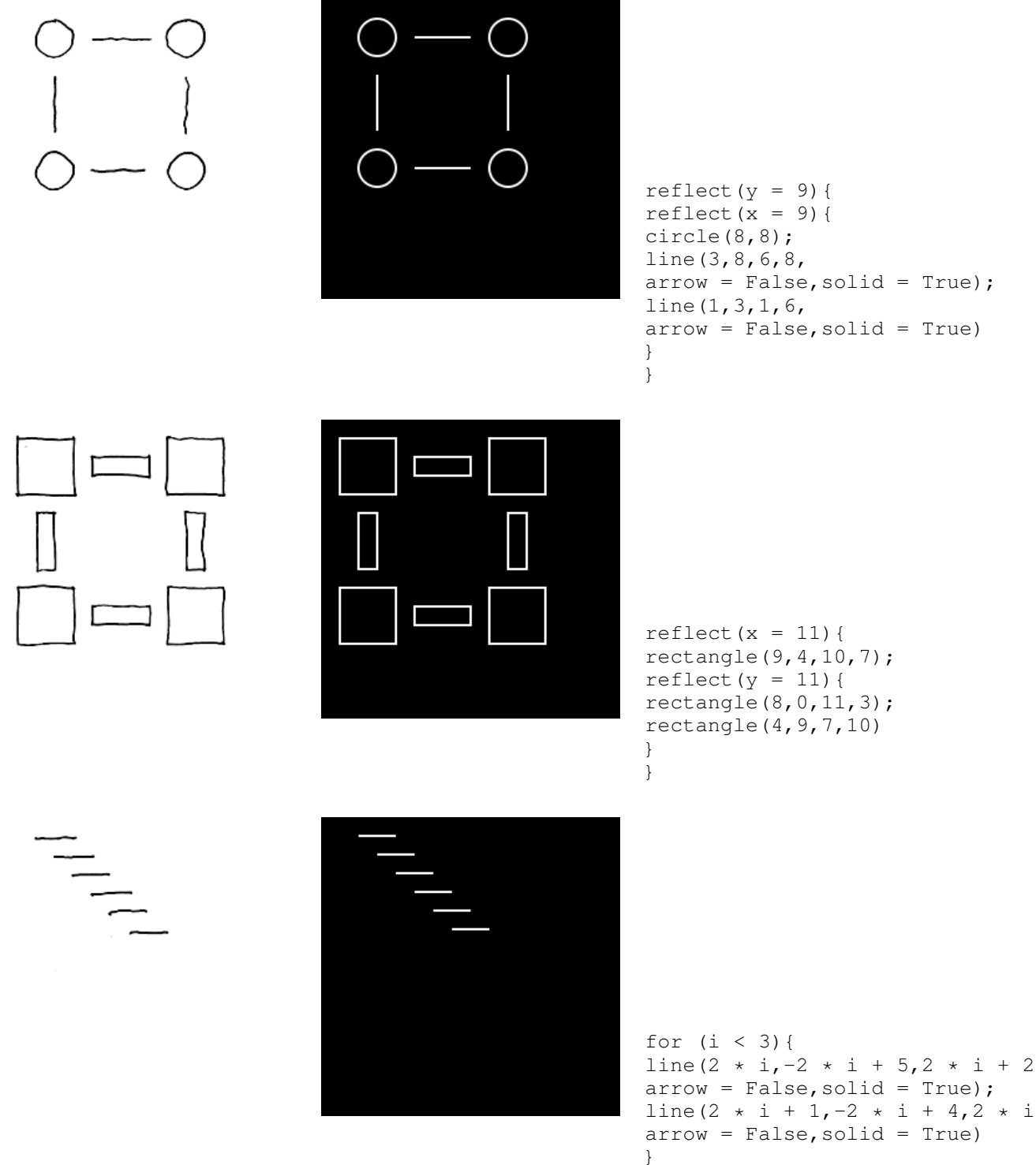

```
reflect(y = 9){
reflect(x = 9){
circle(8,8);
line(3,8,6,8,
arrow = False,solid = True);
line(1,3,1,6,
arrow = False,solid = True)
}
}
```

```
reflect(x = 11){
rectangle(9,4,10,7);
reflect(y = 11){
rectangle(8,0,11,3);
rectangle(4,9,7,10)
}
}
```

```
for (i < 3){
line(2 * i,-2 * i + 5,2 * i + 2,
arrow = False,solid = True);
line(2 * i + 1,-2 * i + 4,2 * i
arrow = False,solid = True)
}
```

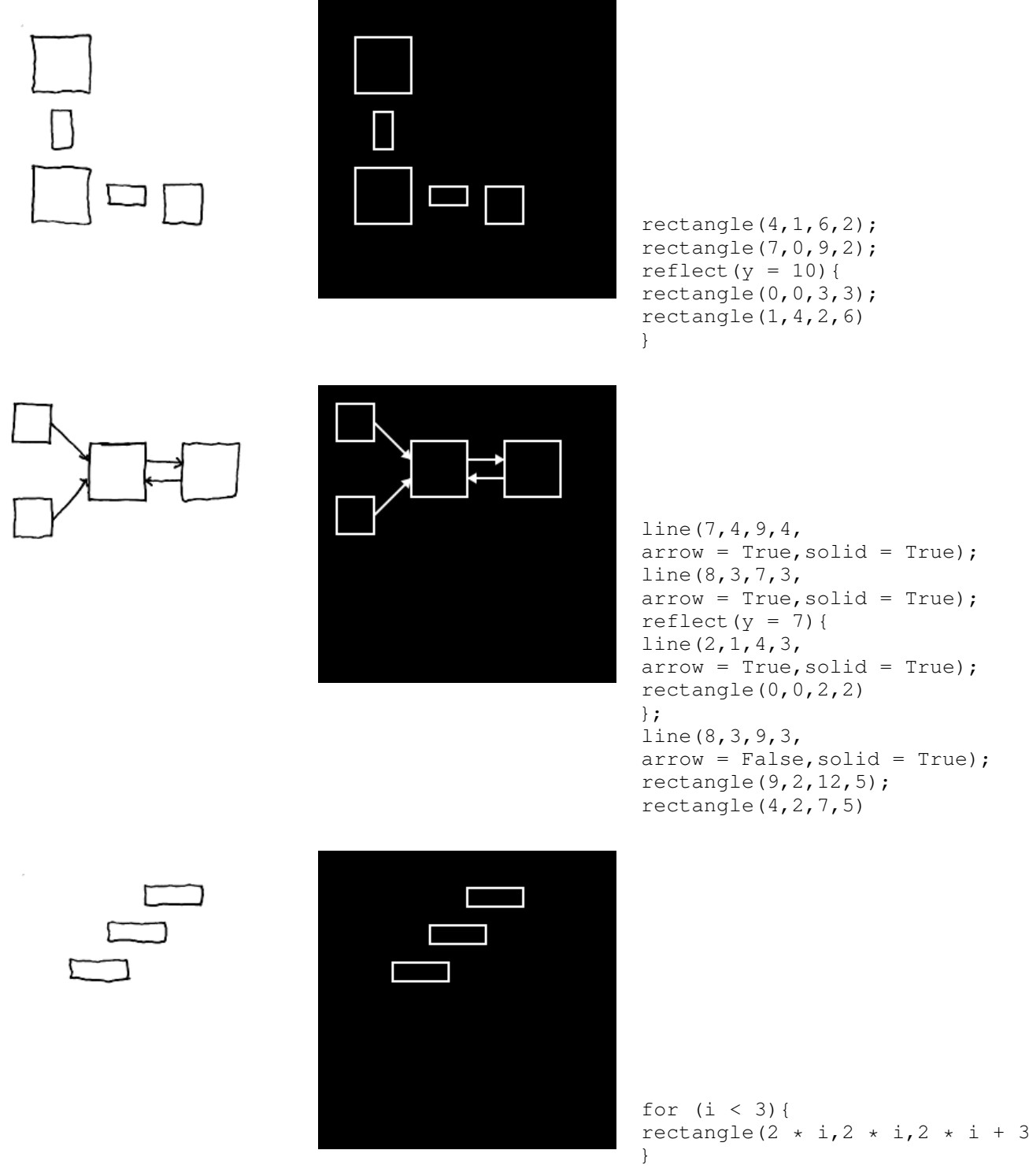

```
rectangle(4,1,6,2);
rectangle(7,0,9,2);
reflect(y = 10){
rectangle(0,0,3,3);
rectangle(1,4,2,6)
}
```

```
line(7,4,9,4,
arrow = True,solid = True);
line(8,3,7,3,
arrow = True,solid = True);
reflect(y = 7){
line(2,1,4,3,
arrow = True,solid = True);
rectangle(0,0,2,2)
};
line(8,3,9,3,
arrow = False,solid = True);
rectangle(9,2,12,5);
rectangle(4,2,7,5)
```

```
for (i < 3){
rectangle(2 * i,2 * i,2 * i + 3,
}
```

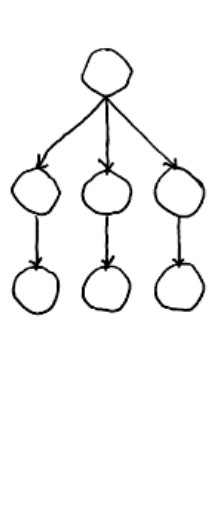
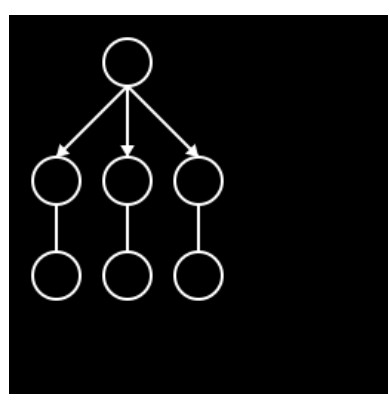

```
circle(4,10);
for (i < 3){
circle(3 * i + 1,1);
circle(3 * i + 1,5);
line(4,9,3 * i + 1,6,
arrow = True,solid = True);
line(3 * i + 1,4,3 * i + 1,2,
arrow = True,solid = True)
}
```

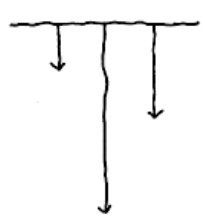
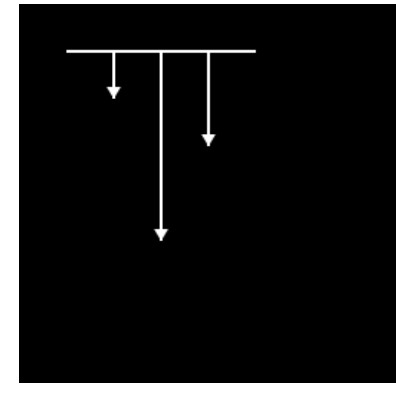

```
line(2,8,2,6,
arrow = True,solid = True);
line(4,8,4,0,
arrow = True,solid = True);
line(6,8,6,4,
arrow = True,solid = True);
line(0,8,8,8,
arrow = False,solid = True)
```

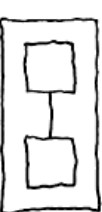
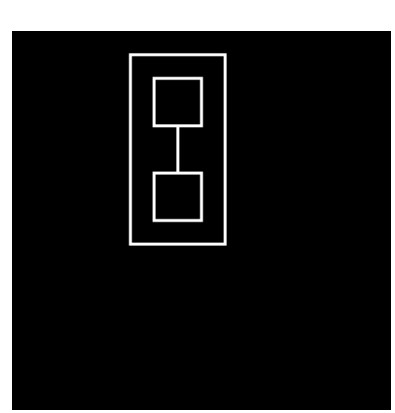

```
line(2,3,2,5,
arrow = False,solid = True);
rectangle(0,0,4,8);
rectangle(1,1,3,3);
rectangle(1,5,3,7)
```

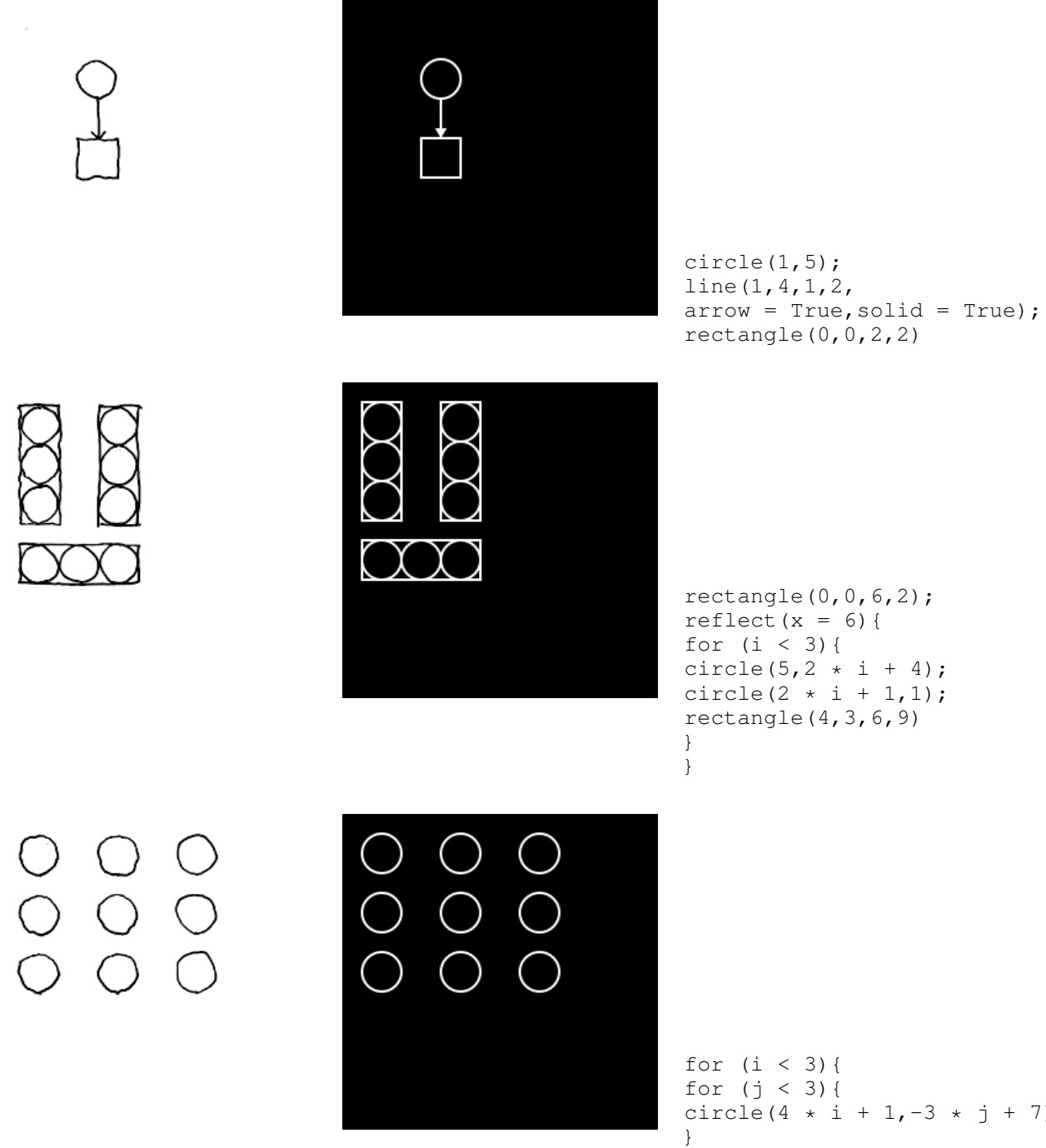

```
circle(1,5);
line(1,4,1,2,
arrow = True,solid = True);
rectangle(0,0,2,2)
```

```
rectangle(0,0,6,2);
reflect(x = 6){
for (i < 3){
circle(5,2 * i + 4);
circle(2 * i + 1,1);
rectangle(4,3,6,9)
}
}
```

```
for (i < 3){
for (j < 3){
circle(4 * i + 1,-3 * j + 7)
}
}
```

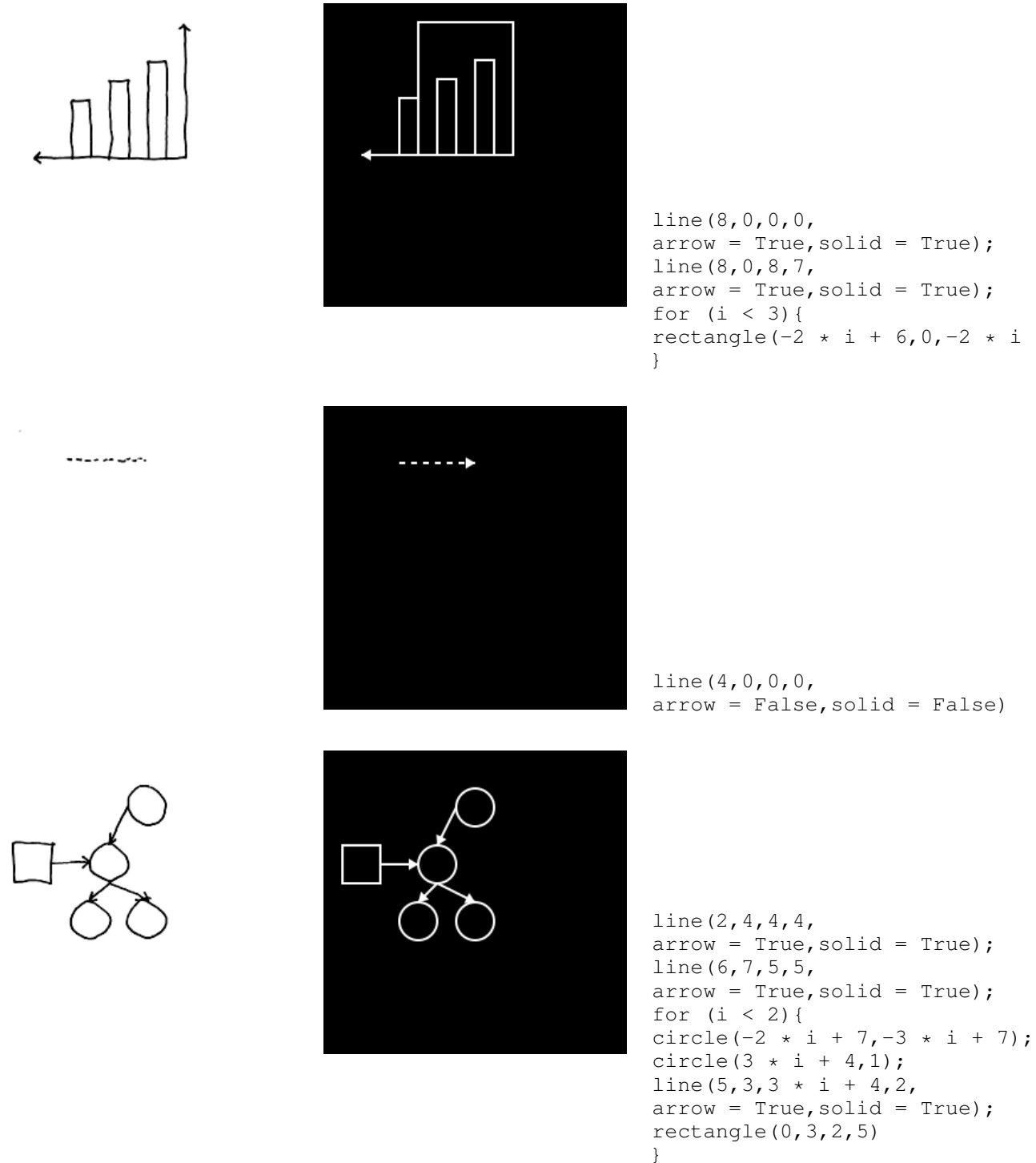

```
line(8,0,0,0,
arrow = True,solid = True);
line(8,0,8,7,
arrow = True,solid = True);
for (i < 3){
rectangle(-2 * i + 6,0,-2 * i +
}
```

```
line(4,0,0,0,
arrow = False,solid = False)
```

```
line(2,4,4,4,
arrow = True,solid = True);
line(6,7,5,5,
arrow = True,solid = True);
for (i < 2){
circle(-2 * i + 7,-3 * i + 7);
circle(3 * i + 4,1);
line(5,3,3 * i + 4,2,
arrow = True,solid = True);
rectangle(0,3,2,5)
}
```

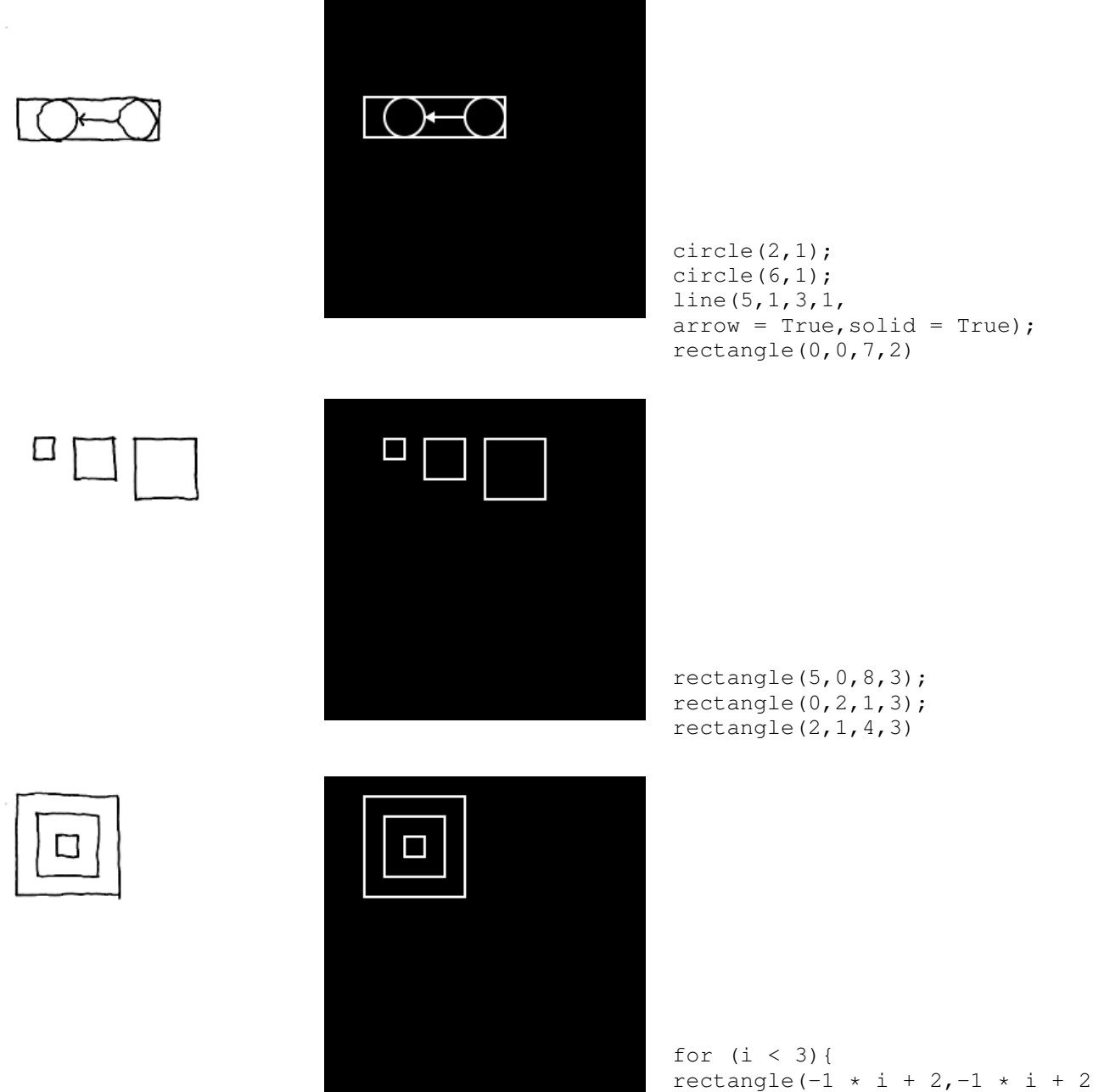

```
circle(2,1);
circle(6,1);
line(5,1,3,1,
arrow = True,solid = True);
rectangle(0,0,7,2)
```

```
rectangle(5,0,8,3);
rectangle(0,2,1,3);
rectangle(2,1,4,3)
```

```
for (i < 3){
rectangle(-1 * i + 2,-1 * i + 2,
}
```

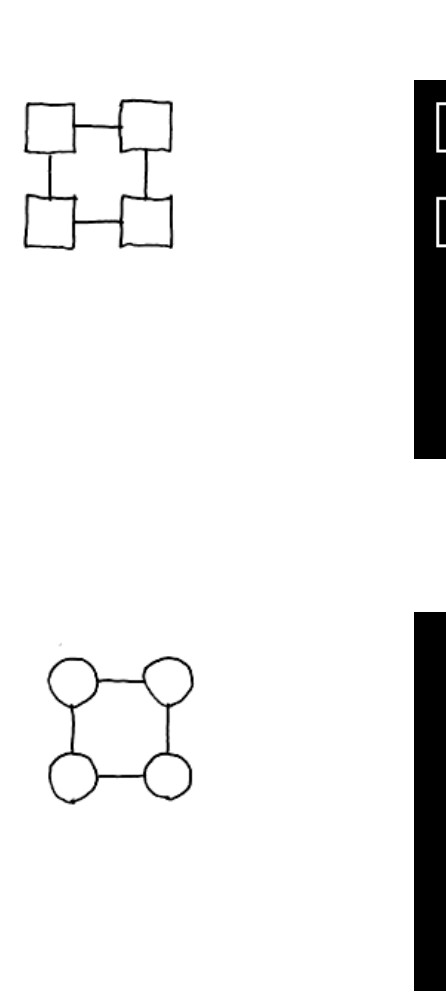

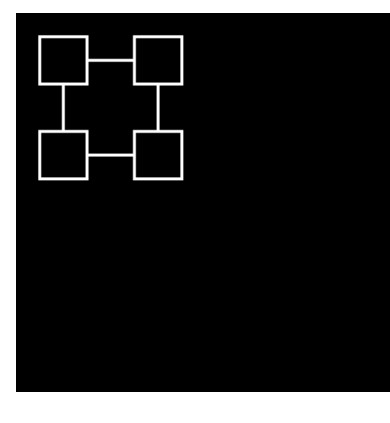

```
reflect(y = 6){
line(2,5,4,5,
arrow = False,solid = True);
reflect(x = 6){
line(5,2,5,4,
arrow = False,solid = True);
rectangle(0,4,2,6)
}
}
```

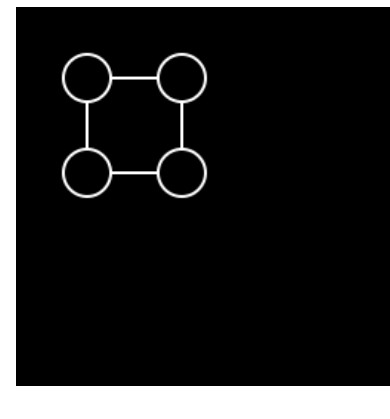

```
reflect(y = 6){
reflect(x = 6){
circle(1,1);
line(5,2,5,4,
arrow = False,solid = True)
};
line(2,1,4,1,
arrow = False,solid = True)
}
```

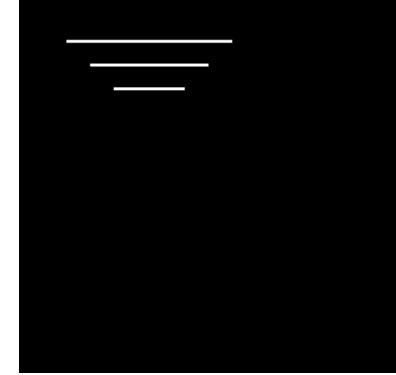

```
for (i < 3){
line(1 * i,-1 * i + 2,-1 * i + 7
arrow = False,solid = True)
}
```

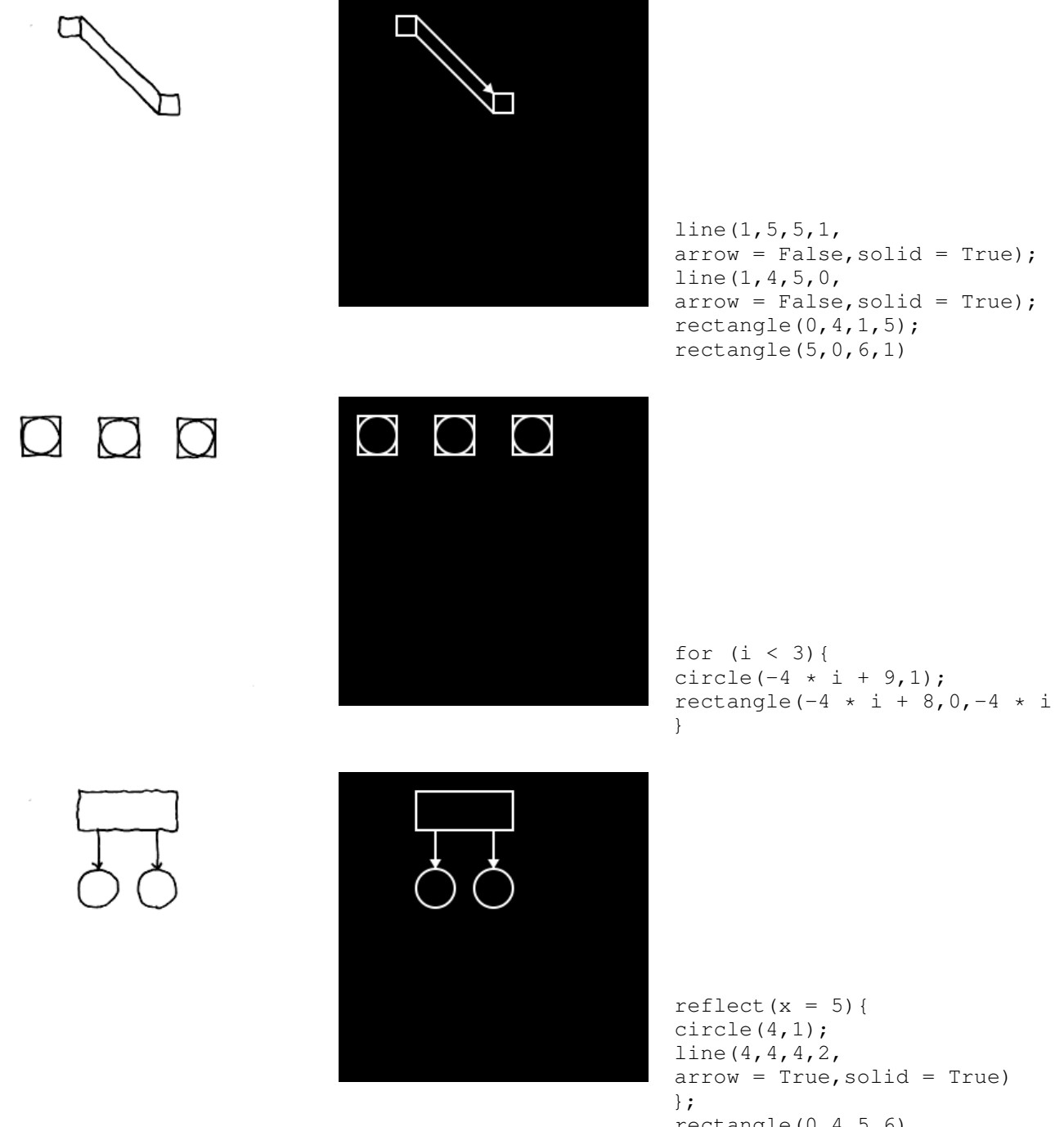

```
line(1,5,5,1,
arrow = False,solid = True);
line(1,4,5,0,
arrow = False,solid = True);
rectangle(0,4,1,5);
rectangle(5,0,6,1)
```

```
for (i < 3){
circle(-4 * i + 9,1);
rectangle(-4 * i + 8,0,-4 * i +
}
```

```
reflect(x = 5){
circle(4,1);
line(4,4,4,2,
arrow = True,solid = True)
};
rectangle(0,4,5,6)
```

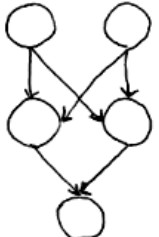 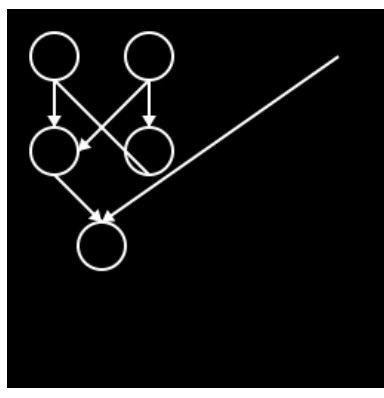

```
circle(3,1);
reflect(x = 6){
circle(5,5);
circle(1,9);
line(5,4,3,2,
arrow = True,solid = True);
line(5,8,2,5,
arrow = True,solid = True);
line(1,8,1,6,
arrow = True,solid = True)
}
```

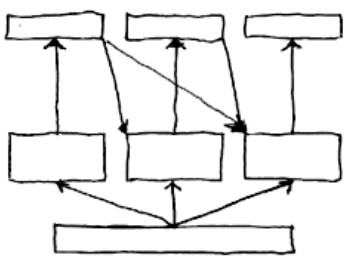 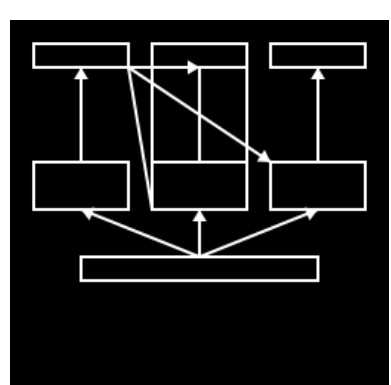

```
for (i < 3){
line(7,1,5 * i + 2,3,
arrow = True,solid = True);
for (j < (1*i + 1)){
if (j > 0){
line(5 * j + -1,9,5 * i,5,
arrow = True,solid = True)
}
line(5 * j + 2,5,5 * j + 2,9,
arrow = True,solid = True)
};
rectangle(5 * i,3,5 * i + 4,5);
rectangle(5 * i,9,5 * i + 4,10)
};
rectangle(2,0,12,1)
```

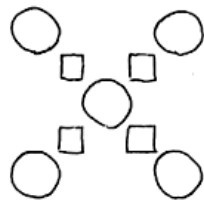 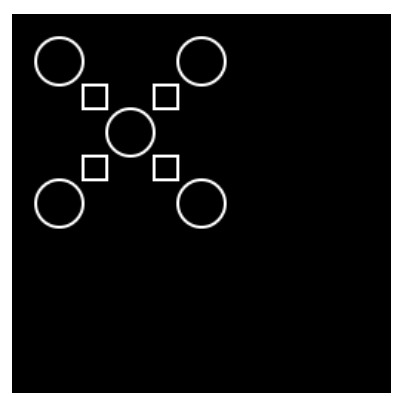

```
reflect(y = 8){
for (i < 3){
circle(-3 * i + 7,-3 * i + 7)
};
rectangle(2,2,3,3);
rectangle(5,5,6,6)
}
```

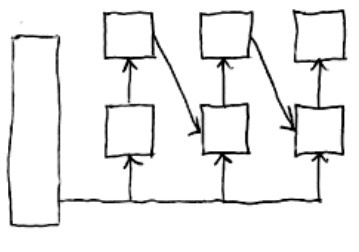 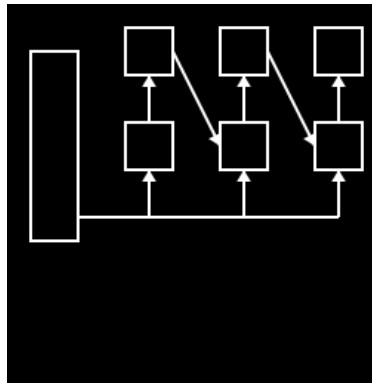

```
line(10,8,12,4,
arrow = True,solid = True);
line(6,8,8,4,
arrow = True,solid = True);
for (i < 3){
line(4 * i + 5,5,4 * i + 5,7,
arrow = True,solid = True);
line(4 * i + 5,1,4 * i + 5,3,
arrow = True,solid = True);
rectangle(4 * i + 4,3,4 * i + 6,
rectangle(4 * i + 4,7,4 * i + 6,
line(2,1,13,1,
arrow = False,solid = True)
};
rectangle(0,0,2,8)
```

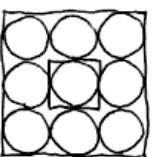 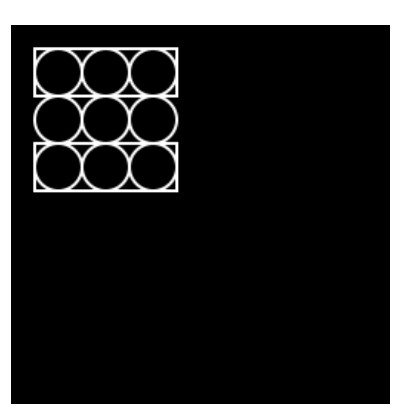

```
for (i < 3){
for (j < 3){
circle(2 * j + 1,2 * i + 1)
}
};
rectangle(2,2,4,4);
rectangle(0,0,6,6)
```

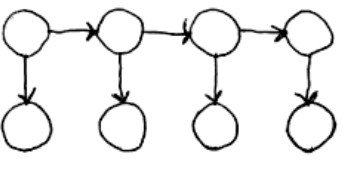 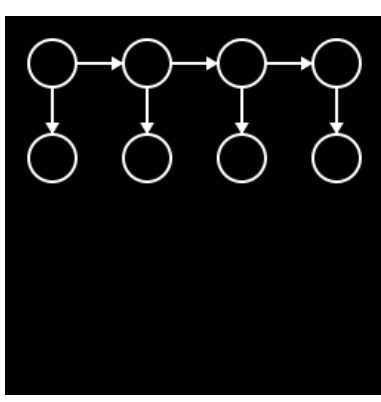

```
for (i < 4){
circle(4 * i + 1,1);
circle(4 * i + 1,5);
for (j < 3){
line(4 * i + 1,4,4 * i + 1,2,
arrow = True,solid = True);
line(4 * j + 2,5,4 * j + 4,5,
arrow = True,solid = True)
}
}
```

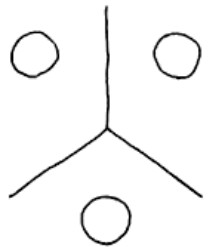 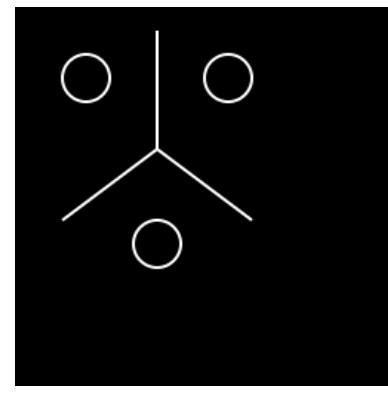

```
reflect(x = 8){
circle(4,1);
circle(1,8);
line(0,2,4,5,
arrow = False,solid = True);
line(4,5,4,10,
arrow = False,solid = True)
}
```

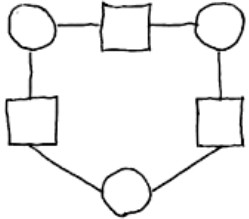 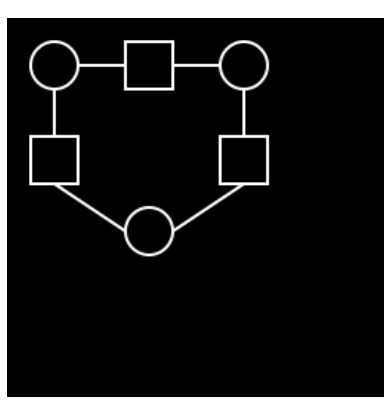

```
circle(9,8);
circle(5,1);
circle(1,8);
reflect(x = 10){
line(6,1,9,3,
arrow = False,solid = True);
line(2,8,4,8,
arrow = False,solid = True);
line(9,5,9,7,
arrow = False,solid = True);
rectangle(0,3,2,5)
};
rectangle(4,7,6,9)
```

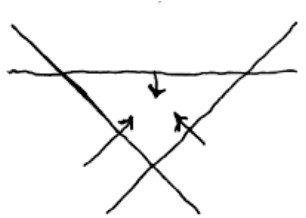 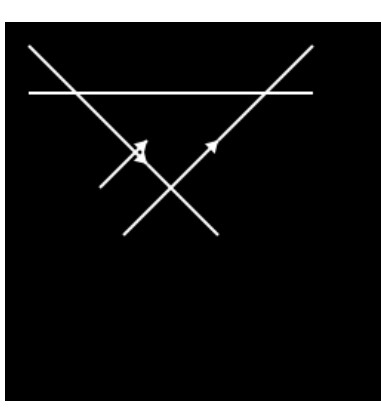

```
line(3,2,5,4,
arrow = True,solid = True);
line(6,6,6,5,
arrow = True,solid = True);
line(8,3,7,4,
arrow = True,solid = True);
line(4,0,12,8,
arrow = False,solid = True);
line(0,6,12,6,
arrow = False,solid = True);
line(0,8,8,0,
arrow = False,solid = True)
```

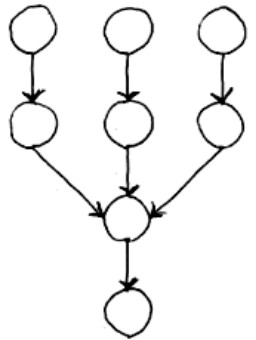 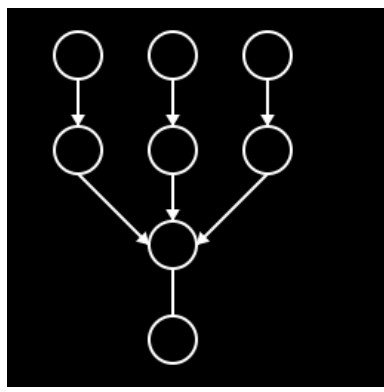

```
for (i < 3){
circle(4 * i + 1,13);
circle(5,-4 * i + 9);
circle(4 * i + 1,9);
line(4 * i + 1,12,4 * i + 1,10,
arrow = True,solid = True);
line(5,-4 * i + 12,5,-4 * i + 10
arrow = True,solid = True)
};
line(9,8,6,5,
arrow = True,solid = True);
line(1,8,4,5,
arrow = True,solid = True)
```

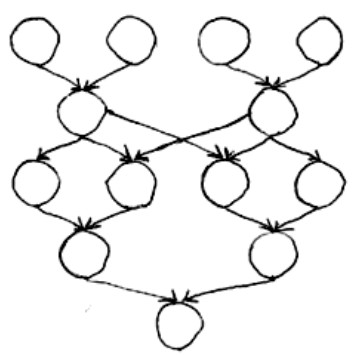 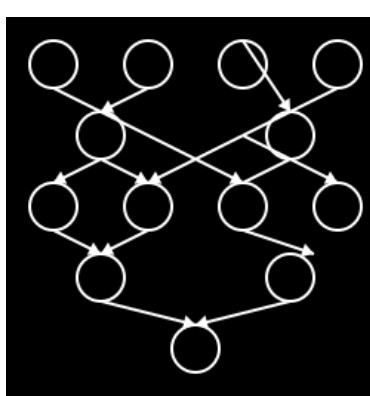

```
reflect(x = 14){
circle(11,10);
circle(3,4);
circle(7,1);
reflect(y = 20){
circle(13,7);
circle(9,7)
};
line(3,3,7,2,
arrow = True,solid = True);
line(10,10,5,8,
arrow = True,solid = True);
reflect(x = 6){
line(5,12,3,11,
arrow = True,solid = True);
line(1,6,3,5,
arrow = True,solid = True);
line(3,9,5,8,
arrow = True,solid = True)
}
}
```

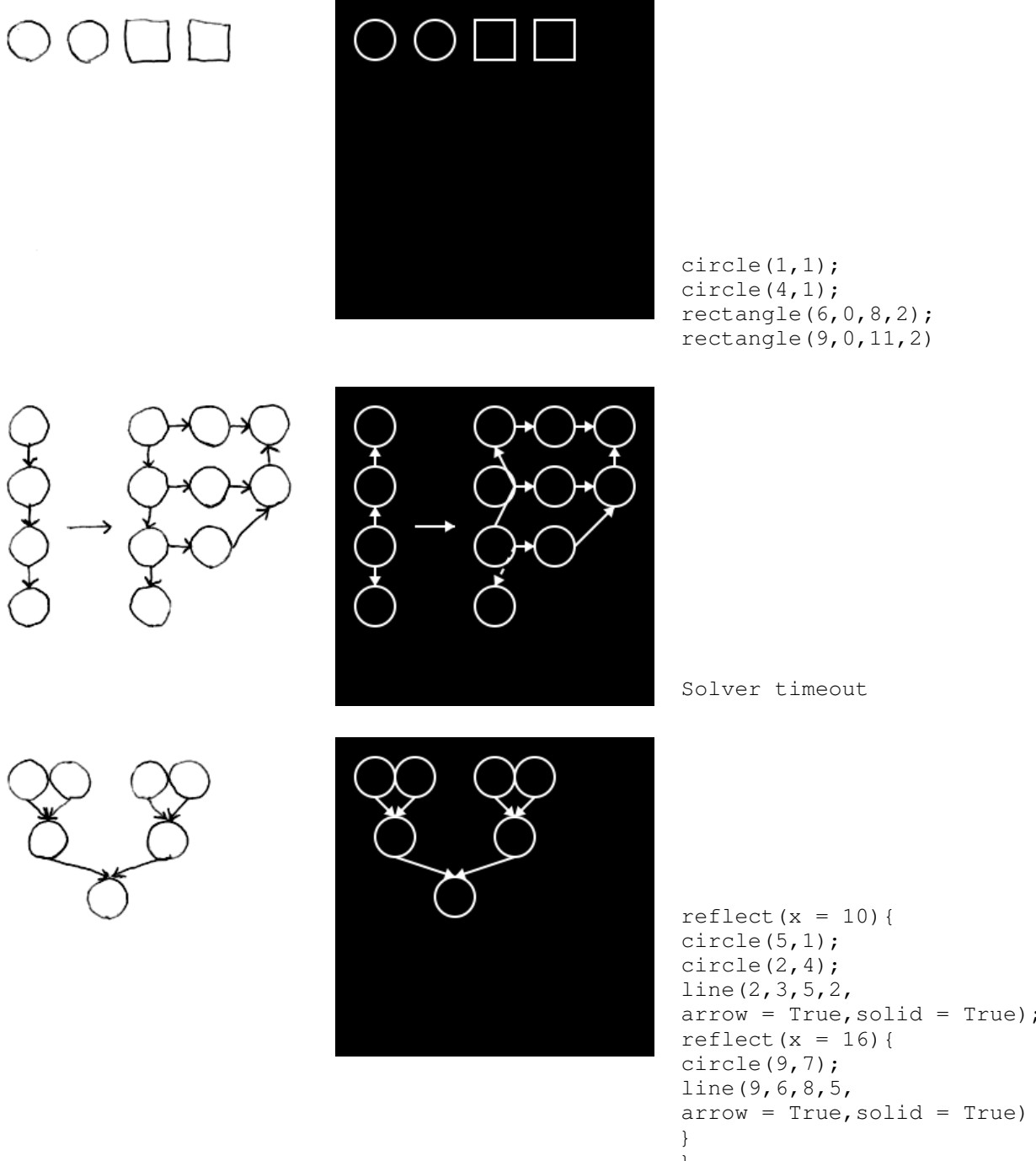

```
circle(1,1);
circle(4,1);
rectangle(6,0,8,2);
rectangle(9,0,11,2)
```

Solver timeout

```
reflect(x = 10){
circle(5,1);
circle(2,4);
line(2,3,5,2,
arrow = True,solid = True);
reflect(x = 16){
circle(9,7);
line(9,6,8,5,
arrow = True,solid = True)
}
}
```

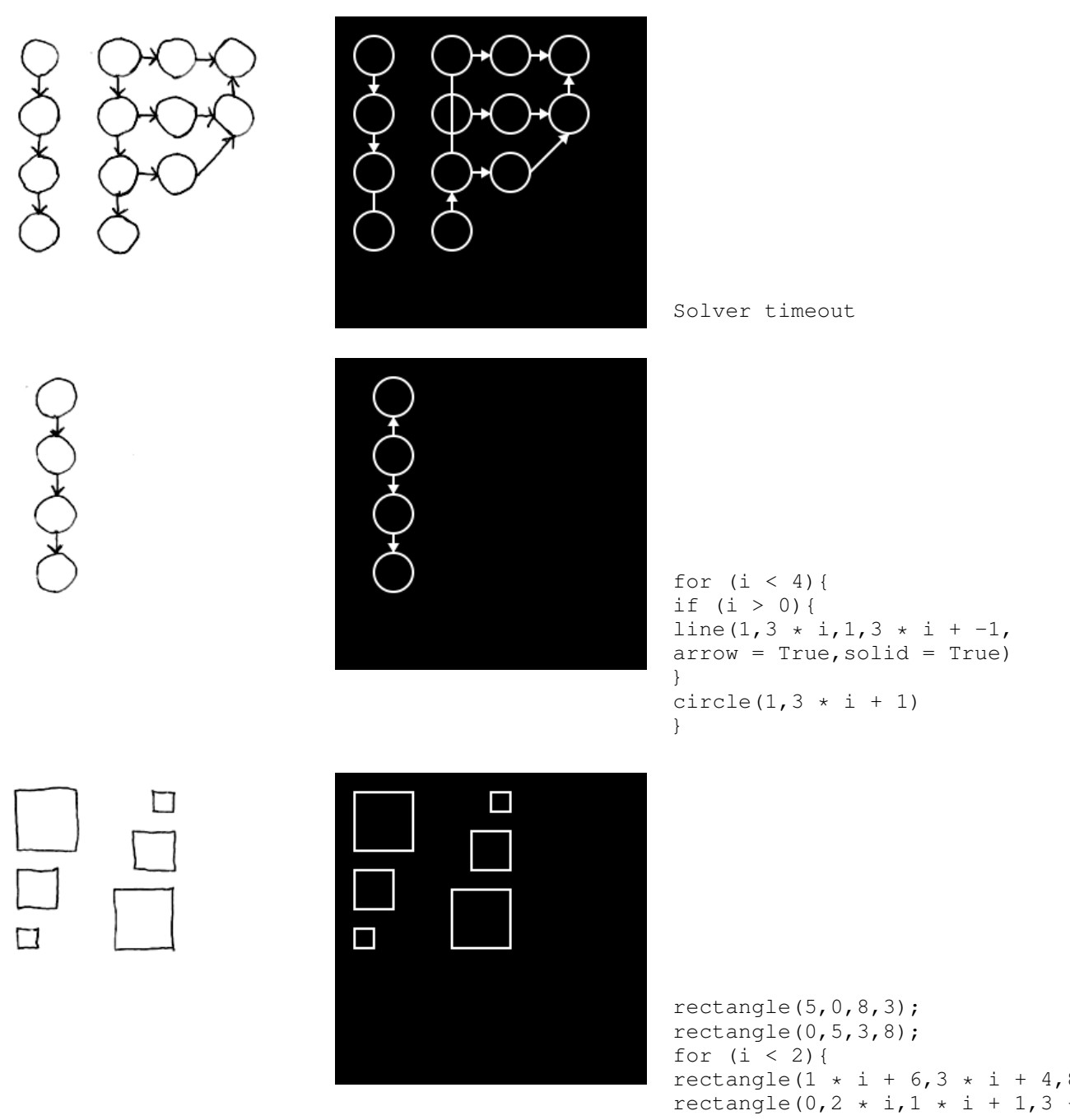

Solver timeout

```
for (i < 4){
if (i > 0){
line(1,3 * i,1,3 * i + -1,
arrow = True,solid = True)
}
circle(1,3 * i + 1)
}
```

```
rectangle(5,0,8,3);
rectangle(0,5,3,8);
for (i < 2){
rectangle(1 * i + 6,3 * i + 4,8,
rectangle(0,2 * i,1 * i + 1,3 *
}
```

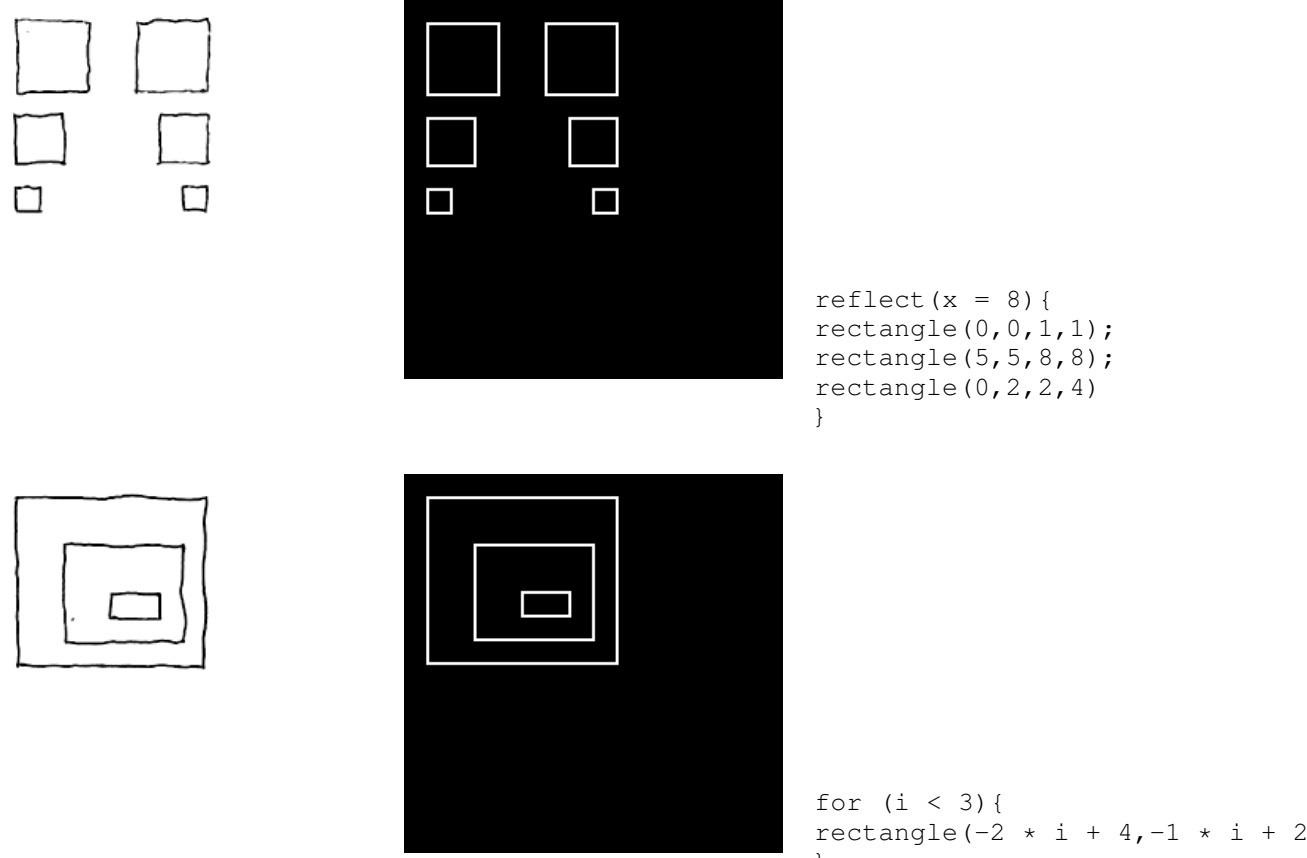

```
reflect(x = 8){
rectangle(0,0,1,1);
rectangle(5,5,8,8);
rectangle(0,2,2,4)
}
```

```
for (i < 3){
rectangle(-2 * i + 4,-1 * i + 2,
}
```

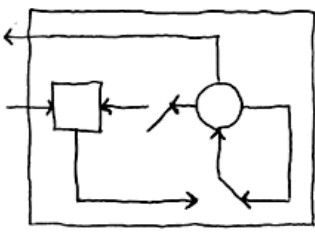
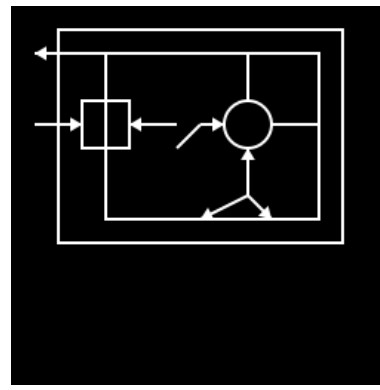

```
circle(9,5);
line(3,1,8,1,
arrow = True,solid = True);
line(8,5,7,5,
arrow = True,solid = True);
line(9,8,0,8,
arrow = True,solid = True);
line(9,2,9,4,
arrow = True,solid = True);
line(12,1,10,1,
arrow = True,solid = True);
line(9,2,10,1,
arrow = False,solid = True);
line(12,1,12,5,
arrow = False,solid = True);
reflect(x = 6){
line(6,5,4,5,
arrow = True,solid = True)
};
rectangle(2,4,4,6);
rectangle(1,0,13,9);
line(9,6,9,8,
arrow = False,solid = True);
line(6,4,7,5,
arrow = False,solid = True);
line(10,5,12,5,
arrow = False,solid = True);
line(3,1,3,4,
arrow = False,solid = True)
```

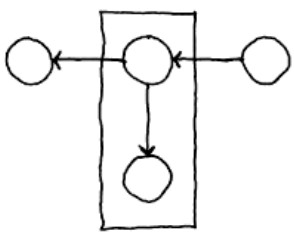
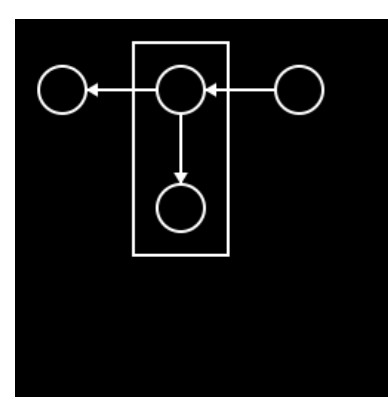

```
circle(6,2);
for (i < 3){
circle(5 * i + 1,7)
};
line(5,7,2,7,
arrow = True,solid = True);
line(6,6,6,3,
arrow = True,solid = True);
line(10,7,7,7,
arrow = True,solid = True);
rectangle(4,0,8,9)
```

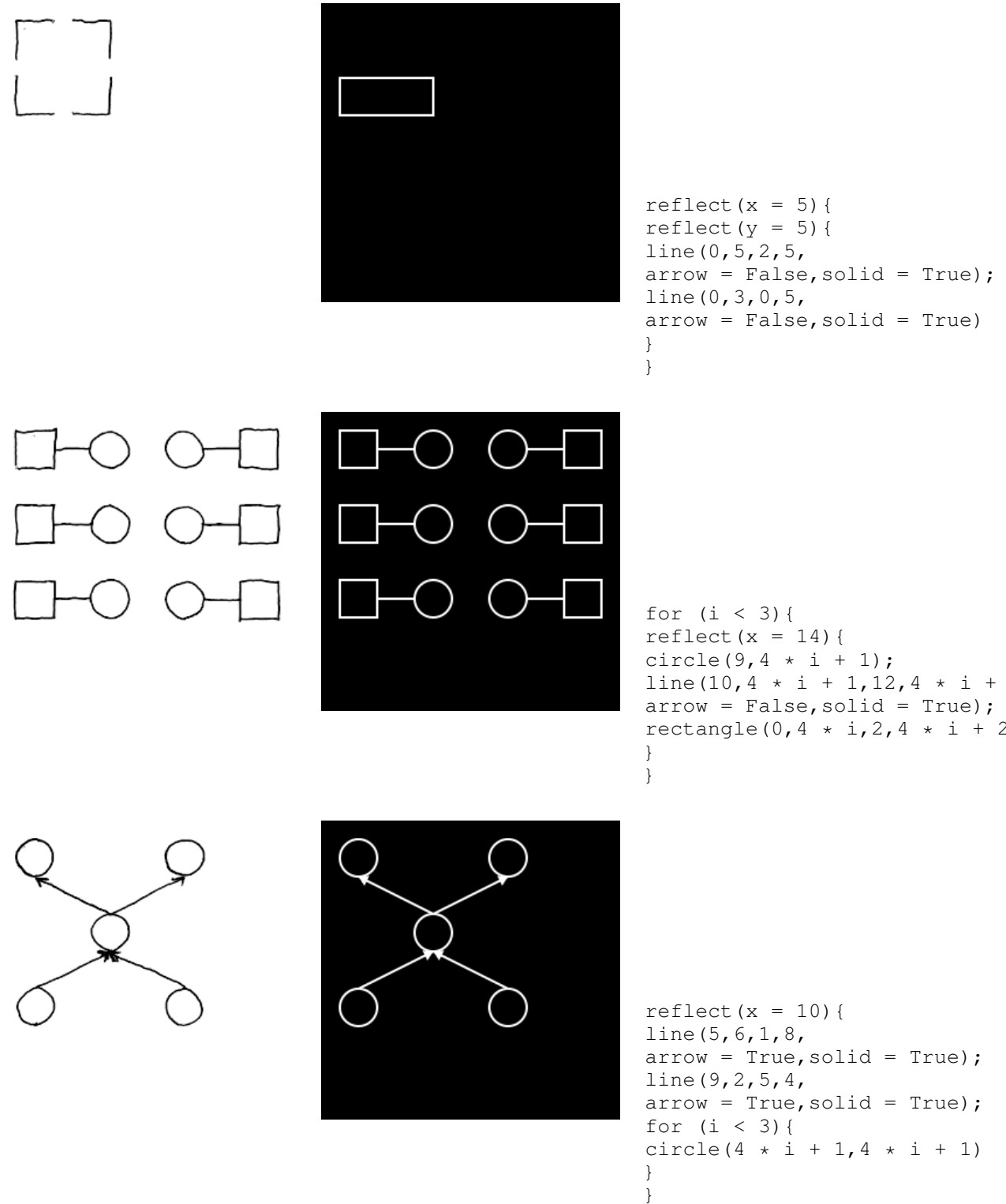

```
reflect(x = 5){
reflect(y = 5){
line(0,5,2,5,
arrow = False,solid = True);
line(0,3,0,5,
arrow = False,solid = True)
}
}
```

```
for (i < 3){
reflect(x = 14){
circle(9,4 * i + 1);
line(10,4 * i + 1,12,4 * i + 1,
arrow = False,solid = True);
rectangle(0,4 * i,2,4 * i + 2)
}
}
```

```
reflect(x = 10){
line(5,6,1,8,
arrow = True,solid = True);
line(9,2,5,4,
arrow = True,solid = True);
for (i < 3){
circle(4 * i + 1,4 * i + 1)
}
}
```

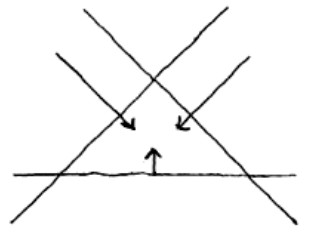
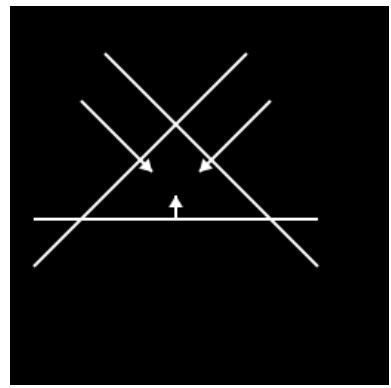

```
reflect(x = 12){
line(6,2,6,3,
arrow = True,solid = True);
line(2,7,5,4,
arrow = True,solid = True);
line(0,0,9,9,
arrow = False,solid = True)
};
line(0,2,12,2,
arrow = False,solid = True)
```

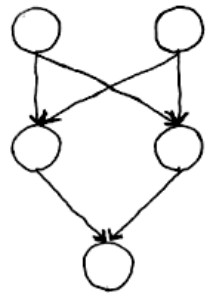
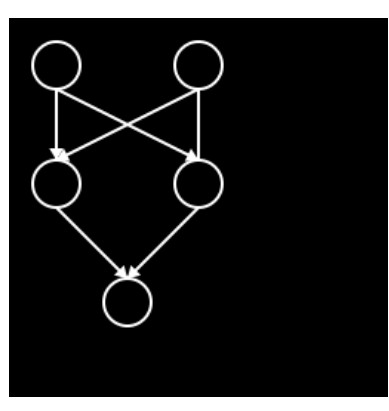

```
for (i < 3){
for (j < 3){
if (j > 0){
circle(6 * i + -5,-5 * j + 16);
line(6 * i + -5,5,4,2,
arrow = True,solid = True);
line(6 * j + -5,10,6 * i + -5,7,
arrow = True,solid = True)
}
circle(4,1)
}
}
```

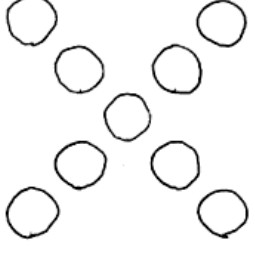
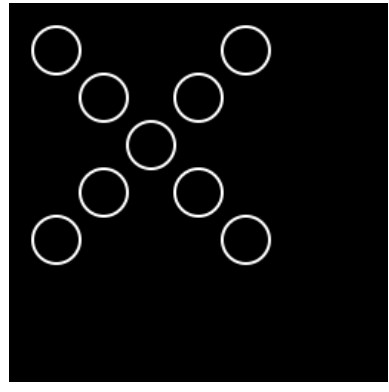

```
reflect(y = 10){
circle(1,9);
for (i < 4){
circle(-2 * i + 9,-2 * i + 9)
}
}
```

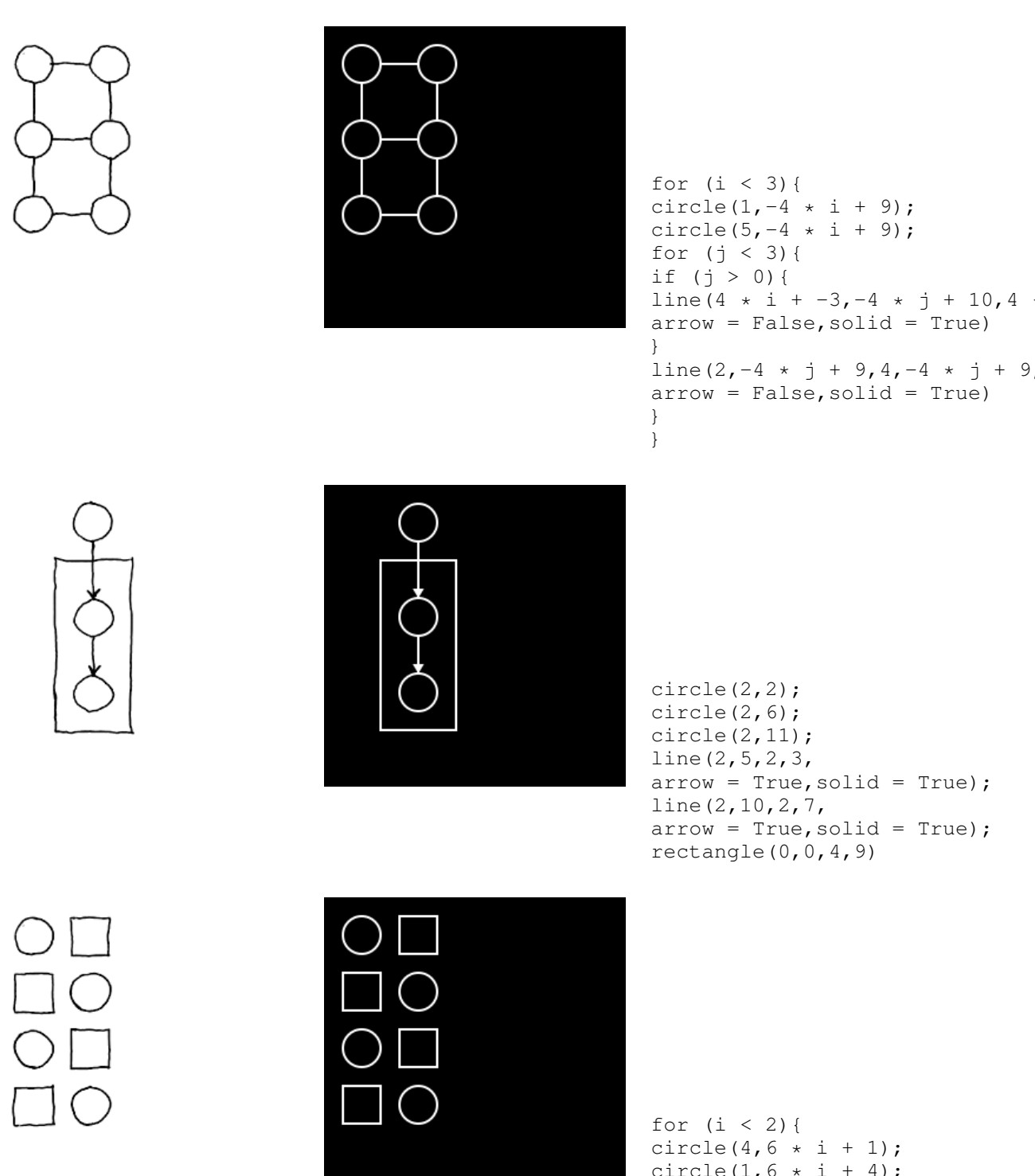

```
for (i < 3){
circle(1,-4 * i + 9);
circle(5,-4 * i + 9);
for (j < 3){
if (j > 0){
line(4 * i + -3,-4 * j + 10,4 *
arrow = False,solid = True)
}
line(2,-4 * j + 9,4,-4 * j + 9,
arrow = False,solid = True)
}
}
```

```
circle(2,2);
circle(2,6);
circle(2,11);
line(2,5,2,3,
arrow = True,solid = True);
line(2,10,2,7,
arrow = True,solid = True);
rectangle(0,0,4,9)
```

```
for (i < 2){
circle(4,6 * i + 1);
circle(1,6 * i + 4);
rectangle(0,6 * i,2,6 * i + 2);
rectangle(3,6 * i + 3,5,6 * i +
}
```

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
