# OpenReview forum: "Learning to Infer Graphics Programs from Hand-Drawn Images"
_ICLR.cc/2018/Conference — Reject_

### Official Review · AnonReviewer2 · 2017-11-28
**A great paper that needs an editing pass**

**Rating:** 4
**Confidence:** 4

**Review:**

I think the idea of inferring programmatic descriptions of handwritten diagrams is really cool, and that the combination of SMC-based inference with constraint-based synthesis is nice. I also think the application is clearly useful – one could imagine that this type of technology would eventually become part of drawing / note-taking applications.

That said, based on the current state of the manuscript, I find it difficult to recommend acceptance. I understand that the ICLR does not strictly have a page limit, but I think submitting a manuscript of over 11 pages is taking things a bit too far. The manuscript would greatly benefit from a thorough editing pass and some judicious reconsideration of space allocated to figures. Moreover, despite its relative verbosity, or perhaps because of it, I found it surprisingly difficult to extract simple implementation details from the text (for example I had to dig up the size of the synthetic training corpus from the 44-page appendix).

Presentation issues aside, I think this is great work. There is a lot here, and I am sympathetic to the challenges of explaining everything clearly in a single (short) paper. That said, I do think that the authors need to take another stab at this to get the manuscript to a point where it can be impactful.

Minor Comments

- I don't understand what the "hypothesis" is in the trace hypothesis. Breaking down the problem into an AIR-style sequential detection task and a program induction is certainly a reasonable thing to do. However, the word "hypothesis" is generally used to refer to a testable explanation of a phenomenon, which is not really applicable here.

- How is the edit distance defined? In particular, are we treating the drawing commands as a set or a sequence when we calculate "the number of drawing commands by which two trace sets differ"?

- I took me a while to understand that the authors first consider the case of SMC for synthetic images with a pixel-based likelihood, and then move on to SMC with and edit-distance based surrogate likelihood for hand-drawn pictures. The text seems to suggest that only 100 of such hand drawn images were actually used, is that correct?

- What does the (+) operator do in Figure 3?

- I am not sure that "correcting errors made by the neural network" is the most accurate way to describe a reranking of the top-k samples returned by the SMC sweep.

- Table 3 is very nice, but does not need to be a full page.

- I would recommend that the authors consolidate wrap-around figures into full-width figures.

---

> ### Author Response · Authors · 2017-12-15
> **Response to anonymous reviewer 3**
>
> Thank you for your helpful review.
>
> We agree that this paper tries to pack a lot of content into one manuscript and would be much improved by an editing pass. Our posted revision is much shorter (9.25 pages, excluding references) and more clearly outlines the content.
>
> We believe one of the reasons why the initial manuscript was difficult to read is because it did not clearly delineate the domain-specific design choices (like the neural architecture and learned distance metric) from the domain-general ideas (the trace hypothesis and the learned search policy). In the posted revision we draw attention to this distinction in the introduction and outline where in the paper each model component is explained.
>
> "The trace hypothesis" is a hypothesis in the sense that it is a claim about how to architect certain AI systems, and the word "hypothesis" is sometimes used for claims like this (for example, "the strong story hypothesis" and the "directed perception hypothesis": see [1]). But we agree that this might be confusing and we are open to renaming it to something like the trace set architecture/framing.
>
> Regarding "How is the edit distance defined?": We treat the drawing commands as a set, and define the edit distance as the size of the symmetric difference between the ground truth set and the set produced by the model.
>
> It is correct and that we evaluate our model on only 100 real hand drawings. These 100 drawings are best thought of as an out-of-sample test set.
>
> Regarding the (+) operator in Figure 3: This is the direct sum operator, which in here takes 2 single-channel images and stacks them to make a single 2-channel image.
>
> References:
> [1] Winston, Patrick Henry. "The Strong Story Hypothesis and the Directed Perception Hypothesis." AAAI Fall Symposium: Advances in Cognitive Systems. 2011.

---

### Official Review · AnonReviewer3 · 2017-11-28
**Nice experiments but a lot of ad-hoc choices**

**Rating:** 6
**Confidence:** 4

**Review:**

Summary of paper:

This paper tackles the problem of inferring graphics programs from hand-drawn images by splitting it into two separate tasks:
(1) inferring trace sets (functions to use in the program) and
(2) program synthesis, using the results from (1).
The usefulness of this split is referred to as the trace hypothesis.

(1) is done by training a neural network on data [input = rendered image; output = trace sets] which is generated synthetically. During test time, a trace set is generated using a population-based method which samples and assigns weights to the guesses made by the neural network based on a similarity metric. Generalization to hand-drawn images is ensured by by learning the similarity metric.

(2) is done by feeding the trace set into a program synthesis tool of Solar Lezama. Since this is too slow, the authors design a search policy which proposes a restriction on the program search space, making it faster. The final loss for (2) in equation 3 takes into consideration the time taken to synthesize images in a search space.

---

Quality: The experiments are thorough and it seems to work. The potential limitation is generalization to non-synthetic data.
Clarity: The high level idea is clear however some of the details are not clear.
Originality: This work is one of the first that tackles the problem described.
Significance: There are many ad-hoc choices made in the paper, making it hard to extract an underlying insight that makes things work. Is it the trace hypothesis? Or is it just that trying enough things made this work?

---

Some questions/comments:
- Regarding the trace set inference, the loss function during training and the subsequent use of SMC during test time is pretty unconventional. The use of the likelihood P_{\theta}[T | I] as a proposal, as the paper also acknowledges, is also unconventional. One way to look at this which could make it less unconventional is to pose the training phase as learning the proposal distribution in an amortized way (instead of maximizing likelihood) as, for example, in [1, 2].
- In section 2.1., the paper talks about learning the surrogate likelihood function L_{learned} in order to work well for actual hand drawings. This presumably stems from the problem of mismatch between the distribution of the synthetic data used for training and the actual hand drawings. But then L_{learned} is also learned from synthetic data. What makes this translate to non-synthetic data? Does this translate to non-synthetic data?
- What does "Intersection over Union" in Figure 8 mean?
- The details for 3.1 are not clear. In particular, what does t(\sigma | T) in equation 3 refer to? Time to synthesize all images in \sigma? Why is the concept of Bias-optimality important?
- It seems from Table 4 that by design, the learned policy for the program search space already limits the search space to programs with maximum depth of the abstract syntax tree of 3. What is the usual depth of an AST when using Sketch?

---

Minor Comments:
- In page 4, section 2.1: "But pixel-wise distance fares poorly... match the model's renders." and "Pixel-wise distance metrics are sensitive... search space over traces." seem to be saying the same thing
- End of page 5: \citep Polozov & Gulwani (2015)
- Page 6: \citep Solar Lezama (2008)

---

References

[1] Paige, B., & Wood, F. (2016). Inference Networks for Sequential Monte Carlo in Graphical Models. In Proceedings of the 33rd International Conference on Machine Learning, JMLR W&CP 48: 3040-3049.
[2] Le, T. A., Baydin, A. G., & Wood, F. (2017). Inference Compilation and Universal Probabilistic Programming. In Proceedings of the 20th International Conference on Artificial Intelligence and Statistics (Vol. 54, pp. 1338–1348). Fort Lauderdale, FL, USA: PMLR.

---

> ### Author Response · Authors · 2017-12-15
> **Response to anonymous reviewer 2**
>
> Thank you for the thoughtful review.
>
> Regarding "The potential limitation is generalization to non-synthetic data": We wish to clarify that, even though the neural network is trained exclusively on synthetic data, we apply it to real hand drawings. We have prominently clarified this in the posted revision.
>
> Thank you for suggesting framing the proposal training as amortized inference. This is a correct and insightful way of communicating the purpose of the neural network, and we have used this framing in the revision.
>
> Regarding "There are many ad-hoc choices made in the paper": Although many of engineering decisions are specific to our domain, we believe that the core generalizable idea is the trace hypothesis, which factors the problem into two independent pieces that can be tackled separately, rather than trying to go straight from raw input to a program. One could also consider using an amortized inference approach that does not use the trace set as an intermediate steppingstone, like RobustFill [1], or other ICLR papers currently under review [2]. There would be two problems with this hypothetical alternative perception->program approach:
> 1. We would need a large data set of (image, program) pairs, where the programs are drawn from the actual distribution that real-world diagrams are drawn from.
> 2. As shown experimentally in DeepCoder [3], neural approaches to program synthesis that attempt to go directly from the problem specification to the program tend to not work as well in practice as those that also leverage symbolic approaches to program synthesis.
> In the posted revision, we have clarified the boundary between the domain-specific design choices and what we believe to be the domain-general ideas.
>
> The reason we need to learn a surrogate likelihood function L_{learned} is not because of the mismatch between the distribution of the synthetic data and the actual hand drawings. Instead, it is because we need to wrap a stochastic search procedure (SMC) around the neurally-guided proposals, and the SMC sampler needs some way of measuring how well a particle explains an image that is robust to variations in the exact details of how something was drawn, which means we can't use pixel-wise distance. L_{learned} generalizes to real data because it is trained on LaTeX TikZ output rendered with the "pencildraw" package, which causes LaTeX output to look like it was drawn with a pencil.
>
> Thank you for pointing out the fact that we did not define the "Intersection over Union" (IoU). The IoU for two sets A and B is $|A\cap B|/|A\cup B|$. We use IoU to measure the system's accuracy at recovering trace sets. Here the sets are sets of primitive drawing commands.
>
> $t(\sigma | T)$ is the length of time it takes the program synthesizer to find the minimum cost program in $\sigma$ such that that program evaluates to the trace set $T$. Our revision now clarifies this point of confusion.
>
> Bias optimality buys us three important things. First, it guarantees that the policy will always eventually find the minimum cost program. Second, it explicitly takes into account the cost of searching, in contrast to e.g. DeepCoder [3]. Lastly it gives us a differentiable loss function for the policy parameters. The posted revision now discusses these points.
>
> You are correct to notice that the program space is already limited to programs with a maximum depth of 3 - meaning that we can have loops within loops, but not loops within loops within loops. Sketch does not support unbounded program spaces. Most of our graphics programs have depth 2-3.
>
> References:
> [1] Devlin, J., Uesato, J., Bhupatiraju, S., Singh, R., Mohamed, A.R. and Kohli, P., RobustFill: Neural Program Learning under Noisy I/O. 2017.
> [2] Neural Program Search: Solving Data Processing Tasks from Description and Examples. Under review at ICLR 2018.
> [3] Balog, M., Gaunt, A. L., Brockschmidt, M., Nowozin, S., & Tarlow, D. Deepcoder: Learning to write programs. ICLR 2017.

---

### Official Review · AnonReviewer1 · 2017-12-02
**Interesting algorithms and results, but too many contents in a single paper**

**Rating:** 4
**Confidence:** 2

**Review:**

This paper proposes a method to infer lines of code that produces a given image. The method consists of two components. One is to generate traces, which are primitive commands of a graphic program, given an image. The other is to infer lines of code given traces. The first component uses a deep neural network for the conversion and a novel architecture is used for the network. The second component uses a learnt search polity to speed up the inference. Experimental results on a small dataset show that the proposed method can generate lines of code of a graphics program for the images reasonably well. It also discusses possible applications of the method.

Overall, the paper is interesting and the proposed method seems reasonable. Also, it is well contrasted with related work. However, the paper contains too many contents and it is hard to understand the important details without reading supplement and the references. It might be even worth considering to split the paper into two ones and each paper proposes one idea (component) at a time with more details.

That said, I understood the basic ideas of the paper and I liked them. My concern is only around how to write.

---

> ### Author Response · Authors · 2017-12-15
> **Response to anonymous reviewer 1**
>
> Thank you for your thoughtful reviewing. We agree that this paper tries to pack a lot of content into one manuscript and would be much improved by an editing pass. Our posted revision is much shorter (9.25 pages, excluding references) and more clearly outlines the content.

---

> > ### Comment · AnonReviewer1 · 2018-01-18
> > **Thank you for the revision. Became better, but still needs more work.**
> >
> > I think the paper became better. However, it still needs more work.
> >
> > Overall, it is not very clear what to be solved in the paper -- if they want to verify the trace hypothesis, or they want to show that the combination of the proposed components is important to build a system for the problem, or the improvement of each component to deal with difficult challenges is important, or all of these.
> >
> > For example, in Section 1, the authors summarize the contributions of the paper. However, right after that, they mention challenges of the paper which do not correspond to the contributions. The contributions are made not by addressing the challenges.
> >
> > I would suggest a major revision to the paper. I think the concept is good and it can be potentially a great paper.
> >
> > Minor comments:
> >
> > Experiment 1 is not very clear to me. Neural network with SMC is explained in Section 2, but other methods are not well explained. Figure 4 implies all methods use "particle" in some ways. I do not know how "particle" is used in beach search for example.
> >
> > Should Section 6 be "Conclusion"?

---

### Decision · Program_Chairs · 2018-01-29
**ICLR 2018 Conference Acceptance Decision**

**Decision:**

Reject

**Comment:**

The paper addresses an interesting problem, is novel and works.
While the paper improved through reviews + rebuttal, the reviewers still find the presentation lacking.